# Multimodal Contrastive Learning with LIMoE: the Language-Image Mixture of Experts

**Basil Mustafa**[*], **Carlos Riquelme**[*], **Joan Puigcerver**[*], **Rodolphe Jenatton, Neil Houlsby**
Google Brain
{basilm, rikel, jpuigcerver, rjenatton, neilhoulsby}@google.com

## Abstract

Large sparsely-activated models have obtained excellent performance in multiple domains. However, such models are typically trained on a single modality at a time. We present the Language-Image MoE, LIMoE, a sparse mixture of experts model capable of multimodal learning. LIMoE accepts both images and text simultaneously, while being trained using a contrastive loss. MoEs are a natural fit for a multimodal backbone, since expert layers can learn an appropriate partitioning of modalities. However, new challenges arise; in particular, training stability and balanced expert utilization, for which we propose an entropy-based regularization scheme. Across multiple scales, we demonstrate remarkable performance improvement over dense models of equivalent computational cost. LIMoE-L/16 trained comparably to CLIP-L/14 achieves 78.6% zero-shot ImageNet accuracy (vs. 76.2%), and when further scaled to H/14 (with additional data) it achieves 84.1%, comparable to state-of-the-art methods which use larger custom per-modality backbones and pre-training schemes. We analyse the quantitative and qualitative behavior of LIMoE, and demonstrate phenomena such as differing treatment of the modalities and the organic emergence of modality-specific experts.

## 1 Introduction

Sparsely activated mixture of expert (MoE) models have recently been used with great effect to scale up both vision [1, 2] and text models [3, 4]. The primary motivation for using MoEs is to scale model parameters while keeping compute costs under control. These models however have other benefits; for example, the sparsity protects against catastrophic forgetting in continual learning [5] and can improve performance for multitask learning [6] by offering a convenient inductive bias.

Given success in each individual domain, and the intuition that sparse models may better handle distinct tasks, we explore the application of MoEs to multimodal modelling. We take the first step in this direction, and study models that process both images and text. In particular, we train a single multimodal architecture that aligns image and text representations via contrastive learning [7].

When using a setup proposed in prior unimodal models [8, 1], we find that feeding multiple modalities to a single architecture leads to new failure modes unique to MoEs. To overcome these, we present a set of *entropy based regularisers* which stabilise training and improve performance. We call the resulting model LIMoE (Language-Image MoE).

We train a range of LIMoE models which significantly outperform compute-matched dense baselines. We scale this up to a large 5.6B parameter LIMoE-H/14, which applies 675M parameters per token. When evaluated zero-shot [7] on ImageNet-2012 [9] it achieves an accuracy of 84.1%, competitive with two-tower models that make use of modality-specific pre-training and feature extractors, and apply 3-4x more parameters per token.

---

[*]Authors contributed equally.

36th Conference on Neural Information Processing Systems (NeurIPS 2022).

In summary, our contributions are as follows.

- We propose `LIMoE`, the first large-scale multimodal mixture of experts models.

- We demonstrate in detail how prior approaches to regularising mixture of experts models fall short for multimodal learning, and propose a new entropy-based regularisation scheme to stabilise training.

- We show that `LIMoE` generalises across architecture scales, with relative improvements in zero-shot ImageNet accuracy ranging from 7% to 13% over equivalent dense models. Scaled further, `LIMoE`-H/14 achieves 84.1% zero-shot ImageNet accuracy, comparable to SOTA contrastive models with per-modality backbones and pre-training.

- Lastly, we present ablations and analysis to understand the model's behavior and our design decisions.

## 2 Multimodal Mixture of Experts

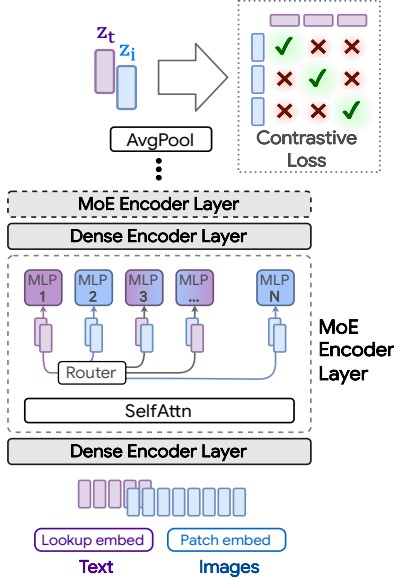

Figure 1: `LIMoE`, a sparsely activated multimodal model, processes both images and texts, utilising conditional computation to allocate computations in a modality-agnostic fashion.

Multimodal contrastive learning typically works with *independent* per-modality encodings [7, 10]. That is, separate models $f_m$ are trained to provide a final representation for every input from the corresponding modality, $m$. In the case of some image and text inputs, $\mathbf{i}$ and $\mathbf{t}$, we have $\mathbf{z_i} = f_{\text{image}}(\mathbf{i})$ and $\mathbf{z_t} = f_{\text{text}}(\mathbf{t})$. For contrastive learning with images and text, this approach results in a "two-tower" architecture, one for each modality. We study a one-tower setup instead, where a *single* model is shared for all modalities, as shown in Figure 1. The one-tower design offers increased generality and scalability, and the potential for cross-modal and cross-task knowledge transfer. We next describe the `LIMoE` architecture and training routine.

### 2.1 Multimodal contrastive learning

Given $n$ pairs of images and text captions $\{(\mathbf{i}_j, \mathbf{t}_j)\}_{j=1}^n$, the model learns representations $\mathcal{Z}_n = \{(\mathbf{z_{i}}_j, \mathbf{z_{t}}_j)\}_{j=1}^n$ such that those corresponding to paired inputs are closer in feature space than those of unpaired inputs. The contrastive training objective [7, 11], with learned temperature $T$, is:

$$\mathcal{L}_j(\mathcal{Z}_n) = \underbrace{-\frac{1}{2} \log \frac{e^{\langle \mathbf{z_{i}}_j, \mathbf{z_{t}}_j \rangle / T}}{\sum_{k=1}^n e^{\langle \mathbf{z_{i}}_j, \mathbf{z_{t}}_k \rangle / T}}}_{\text{image-to-text loss}} \underbrace{-\frac{1}{2} \log \frac{e^{\langle \mathbf{z_{i}}_j, \mathbf{z_{t}}_j \rangle / T}}{\sum_{k=1}^n e^{\langle \mathbf{z_{i}}_k, \mathbf{z_{t}}_j \rangle / T}}}_{\text{text-to-image loss}} . \tag{1}$$

### 2.2 The `LIMoE` Architecture

We use a single Transformer-based architecture for both image and text modalities. The model uses a linear layer per modality to project the intrinsic data dimension to the desired width: for text, a standard one-hot sentencepiece encoding and learned vocabulary [12], and for images, ViT-style patch-based embeddings [13]. Then all tokens are processed by a shared transformer encoder, which is not explicitly conditioned on modality. The token representations from the final layer are average-pooled to produce a single representation vector $\mathbf{z}_m$ for each modality. To compute the training loss in (1), the paired image and text representations are then linearly projected using per-modality weight matrices $\mathbf{W}_m$'s and $\mathcal{L}_j$ is applied to $\{(\mathbf{W}_{\text{image}} \, \mathbf{z_{i}}_k, \mathbf{W}_{\text{text}} \, \mathbf{z_{t}}_k)\}_{k=1}^n$.

This one-tower setup can be implemented with a standard dense Transformer (and we train many such models as baselines). Next, we describe how we introduce MoEs to this setup for `LIMoE`.

**Sparse MoE backbone:** Sparse MoE layers are introduced following the architectural design of [1, 3]. The *experts*—parts of the model activated in an input-dependent fashion—are MLPs. `LIMoE` contains multiple MoE layers. In those layers, each token $\mathbf{x} \in \mathbb{R}^D$ is processed sparsely by $K$ out of $E$ available experts. To choose which $K$, a lightweight router predicts the gating weights *per token*:

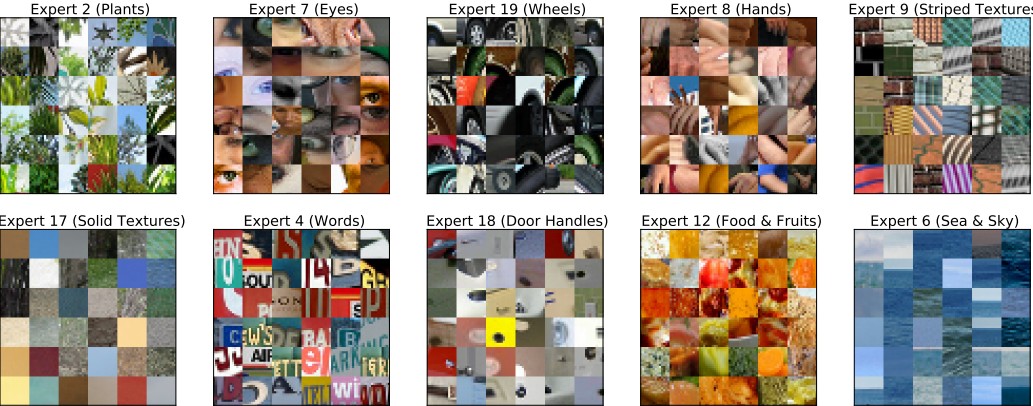

Figure 2: **Token routing examples for Coco.** Image examples of how patches are routed at the MoE layer placed in the 18-th encoder block –i.e. middle of the network– for the `LIMoE-H/14` model.

$g(\mathbf{x}) = \texttt{softmax}(\mathbf{W}_g\mathbf{x}) \in \mathbb{R}^E$ with learned $\mathbf{W}_g \in \mathbb{R}^{D \times E}$. The outputs of the $K$ activated experts are linearly combined according to the gating weights: $\texttt{MoE}(\mathbf{x}) = \sum_{e=1}^{K} g(\mathbf{x})_e \cdot \texttt{MLP}_e(\mathbf{x})$.

Note that, for computational efficiency and implementation constraints, experts have a *fixed buffer capacity*. The number of tokens each expert can process is fixed in advance, and typically assumes that tokens are roughly balanced across experts. If capacity is exceeded, some tokens are "dropped"; they are not processed by the expert, and the expert output is all zeros for those tokens. The rate at which tokens are successfully processed (that is, not dropped) is referred to as the "success rate". It is an important indicator of healthy and balanced routing and often indicative of training stability.

We discovered that routing with tokens from multiple modalities introduces new failure modes; in the next sections we demonstrate this phenomenon, and describe our techniques to address it.

### 2.2.1 Challenges for multimodal MoEs

As mentioned, experts have a fixed buffer capacity. Without intervention, Top-$K$ MoEs tend to "collapse", thus using only one expert. This causes most tokens to be dropped and leads to poor performance [14]. Prior works therefore use auxiliary losses to encourage balanced routing [1, 3, 8].

In multimodal settings, new challenges arise; one is modality misbalance. In realistic setups, there will likely be more of one data type than another. Accordingly, we do not assume or enforce balanced data across modalities, and our experiments have $3-17\times$ more image tokens than text tokens.

Modality-specific experts tend to emerge naturally. In this imbalanced context, this leads to a scenario where all of the tokens from the minority modality get assigned to a single expert, which runs out of capacity. On a global level, routing still appears balanced: tokens from the majority modality are nicely distributed across experts, thereby satisfying modality-agnostic auxiliary losses. For example, in our standard B/16 setup, the router can optimize the importance loss [14] to within 0.5% of its minimum value by perfectly balancing image tokens but dropping all text tokens. This however leads to unstable training and unperforming models.

### 2.2.2 Auxiliary losses

We refer to auxiliary losses used in V-MoE [1] as the *classic* auxiliary losses. We find that they do not yield stable and performant multimodal MoE models. Therefore, we introduce two new losses: the *local entropy loss* and the *global entropy loss*, which are applied on a per-modality basis. We combine these losses with the classic losses; see Appendix B for a summary of all auxiliary losses.

**Definition.** In each MoE layer, for each modality $m$, the router computes a gating matrix $\mathbf{G}_m \in \mathbb{R}^{n_m \times E}$. Each row of $\mathbf{G}_m$ represents the probability distribution over $E$ experts for one of the $n_m$ tokens of that modality in the batch. For a token $\mathbf{x}$ that corresponding row is $p_m(\texttt{experts}|\mathbf{x}) \in \mathbb{R}^E$;

this later dictates which experts process $\mathbf{x}$. The local and global entropy losses are defined by:

$$\Omega_{\text{local}}(\mathbf{G}_m) := \frac{1}{n_m} \sum_{i=1}^{n_m} \mathcal{H}(p_m(\texttt{experts}|\mathbf{x}_i)) \ \text{ and } \ \Omega_{\text{global}}(\mathbf{G}_m) := -\mathcal{H}(\tilde{p}_m(\texttt{experts})), \quad (2)$$

where $\tilde{p}_m(\texttt{experts}) = \frac{1}{n_m} \sum_{i=1}^{n_m} p_m(\texttt{experts}|\mathbf{x}_i)$ is the expert probability distribution averaged over the tokens and $\mathcal{H}(p) = -\sum_{e=1}^{E} p_e \log(p_e)$ denotes the entropy. Note that $\tilde{p}_m(\texttt{experts}) \approx p_m(\texttt{experts})$ since we approximate the true marginal from the tokens in the batch. We use the terminology *local* vs. *global* to emphasise the fact that $\Omega_{\text{local}}$ applies the entropy *locally* for each token while $\Omega_{\text{global}}$ applies the entropy *globally* after having marginalized out the tokens.

**Effects of the losses.** Figure 3 shows why these losses are necessary. With the default losses, modality-specific experts naturally emerge, but the router often changes its preference. This results in unstable training and poor success rate, particularly for the text modality. The local entropy loss encourages concentrated router weights ($p_{\text{text}}(\texttt{experts}|\mathbf{x}_i)$'s have low entropy), but at the expense of the *diversity* of the text experts: the same expert is used for all text tokens (the marginal $\tilde{p}_{\text{text}}(\texttt{experts})$ also has low entropy), leading to dropping. In this setup, many layers have poor text success rates.

To address this, $\Omega_{\text{global}}$ encourages maximization of the marginal entropy, thus pushing $\tilde{p}_{\text{text}}(\texttt{experts})$ towards a more uniform expert distribution. The result is diverse expert usage, stable and confident routing, and high success rates. These are consequently the most performant models.

Intuitively, it is desirable for text tokens to use multiple experts, but not all of them. In order to allow flexibility, we threshold the global entropy loss as $\Omega_{\text{global}}^{\tau}(\mathbf{G}_m) = \max\{0, \tau + \Omega^{\text{global}}(\mathbf{G}_m)\}$, such that the model is encouraged to have a certain minimum entropy, but after exceeding that, the loss is not applied. This avoids distributional collapse but does not apply overly restrictive priors on the routing distribution, as there are many optimal solutions. This can be thought of as a "soft minimum" $S$. With $\tau = \log(S)$, the model must use at least $S$ experts to minimize the loss (either a uniform distribution across $S$ experts -with entropy $\log(S)$-, or a non-uniform distribution using more than $S$). Figure 3b shows the latter occurs; the empirical effect of these thresholds is analysed in Section 4.1.

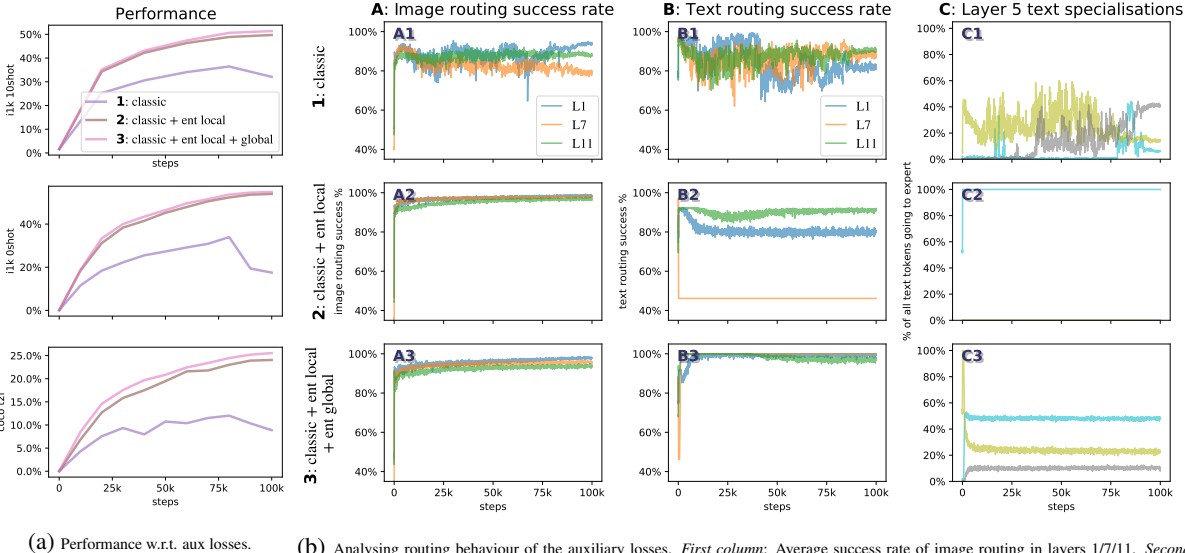

(a) Performance w.r.t. aux losses.

(b) Analysing routing behaviour of the auxiliary losses. *First column*: Average success rate of image routing in layers 1/7/11. *Second column*: Same, for text. *Third column*: In some experts of layer 5, what fraction of all text tokens go to those experts

Figure 3: **What necessitates entropy losses?** *Classic* refers to the standard formulation (importance + load losses [1]). We add the local entropy loss to text tokens (middle row), followed by the global entropy loss (bottom row). **Left:** The "classic" setting is low-performing and unstable. **Right:** Analyzing the entropies shows us why: Without the local loss, the model is prone to unstable changes in expert preferences (C1), and routing success rates are low (A1, B1). The local loss fixes this but causes distributional collapse for one modality (C2), with all text tokens going to one expert (expert 11); this causes even poorer text success rates (B2). This is addressed by the global loss, which has stable expert allocations (C3) and consistently high success rates (A3, B3).

**Connection with mutual information.** The sum $\Omega_{\text{local}}(\mathbf{G}_m) + \Omega_{\text{global}}(\mathbf{G}_m)$ corresponds to the (negative) mutual information [15] between experts and tokens, conditioned on the modality $m$, which we write $-\text{MI}_m(\texttt{experts}; \mathbf{x})$. For each modality taken separately, we are effectively encouraging the knowledge of the token representation to reduce the uncertainty about the experts selection. We also tried other variants of the losses which exploit this connection, such as the mutual information between the experts and modalities, $-\text{MI}(\texttt{experts}; m)$, obtained by first marginalizing the tokens.

### 2.2.3 Priority routing

With Top-$K$ routing, some token dropping is virtually inevitable. Batch Priority Routing (BPR) [1] actively decides which tokens to skip based on their routing weights. It assumes that tokens with a large routing weight are likely to be informative, and should be favored. BPR was mostly used at inference time in [1], allowing for smaller expert capacity buffers. In this setup, one must take care not to systematically favor one modality over the other, for instance, by determining which token to drop based on their rank in the batch, which are usually grouped according to the token modality. BPR provides an essential stabilisation effect during training (Figure 6); we show that it does not trivially rank one modality over another, and it cannot be replaced by other methods of re-ordering the batch. In the appendix we further show how routing priorities compare across text and images.

## 3 Experiments

We study `LIMoE` in the context of multimodal contrastive learning. We first perform a controlled comparison of `LIMoE` to an equivalent "standard" dense Transformer, across a range of model sizes. We then show that when scaled up `LIMoE` can reach a high level of performance. Finally, we ablate the various design decisions leading to `LIMoE` in Section 4.

**Training data.** By default, all models are trained on paired image-text data used in [16], consisting of 3.6B images and alt-texts scraped from the web. For large `LIMoE-H/14` experiment, we also co-train with JFT-4B [17]. We construct artificial text captions from JFT by comma-delimited concatenation of the class names [18]. Appendix A contains full details of our training setup.

**Evaluation.** Our main evaluation is "zero-shot": the model uses its text representations of the classes to make predictions on a new task without extra training data [19, 7]. We focus on image classification accuracy on ImageNet [9] and cross-modal retrieval on MS-COCO [20], following the protocol in [16]. We also evaluate `LIMoE`'s image representations via a linear adaptation protocol [13], and report 10-shot accuracy on ImageNet accuracy accordingly. Where ranges are given, they report 95% confidence intervals across three trials.

### 3.1 Controlled study across scales

We train a range of `LIMoE` models at batch size 16k for 781k steps. This matches the number of training examples used for CLIP [7]. Due to use of different training data and additional tricks, a direct comparison is difficult; we therefore train dense one-tower models as baselines. All models activate $k = 1$ experts per token, similar to Switch Transformer [8].

Figure 4 shows the performance of each model (dense and sparse) against forward-pass FLOPs (for step times and further discussion on compute costs, see Appendix D.2.). The cost-performance Pareto frontier for `LIMoE` dominates the dense models by a wide margin, indicating that `LIMoE` offers strong improvements across all scales from S/32 , up to L/16. The effect is particularly large on zero-shot and 10-shot ImageNet classification, with absolute performance improvements of **10.1%** and **12.2%** on average. For text-to-image retrieval on COCO, `LIMoE` offers a strong boost at small scales, while at larger scales the gains are more modest but still significant.

### 3.2 Scaling up `LIMoE`

We increase the architecture size, training duration, and data size to assess the performance of `LIMoE` in the large-scale regime. In particular, we train a 32-layer `LIMoE-H/14` with 12 expert layers; these are non-uniformly distributed, with 32 experts per layer, and $K = 1$ activated per token. It was trained at a batch size of 21k, introducing 25% JFT-4B images [17] into each batch (with class names as texts). We average checkpoints towards the end of training [21]; refer to Appendix A.3 for details.

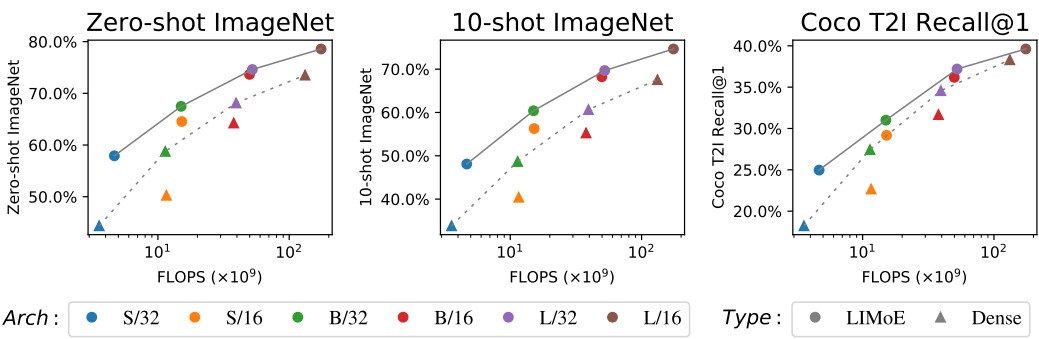

Figure 4: LIMoE scales well to large models, with consistent performance improvements.

The model contains 5.6B parameters in total, but only applies 675M parameters per token. All routers combined account for less than 0.5M parameters. Table 1 shows its performance alongside current state-of-the-art contrastive models. LIMoE achieves 84.1% zero-shot ImageNet classification accuracy with a comparably modest architecture size and training counts. LIMoE is fully trained from scratch, without any pre-trained components, and is the first competitive model with a shared backbone.

In light of its modality agnostic approach, this result is surprisingly strong. Large models handling dozens of distinct tasks are increasingly popular [22], but do not yet approach the state-of-the-art in these tasks. We believe the ability to build a generalist model with specialist components, which can decide how different modalities or tasks should interact, will be key to creating truly multimodal multitask models which excel at everything they do. LIMoE is a promising first step in that direction.

Table 1: Comparing state of the art zero-shot classification models. At a relatively modest scale, LIMoE-H/14 is comparable with the best two-tower models, and it is the first performant one-tower model at this scale. T-x refers to a Transformer [23] with the equivalent parameters of ViT-x [13].

*Key:* [*] *Pretrained*    [PT] *Examples seen during pretraining*    [†] *Uses FixRes [24]*    [§] *Other non-contrastive training objective*

| | Architecture | | Batch | Examples seen | Parameters | ImageNet top-1 % | | | |
|---|---|---|---|---|---|---|---|---|---|
| | Image | Text | size | | per token | Test | V2 | R | A |
| COCA[§] [25] | ViT-g | T-g | 65k | 32.8B | 1.1B | 86.3 | 80.7 | 96.5 | 90.2 |
| BASIC [18] | CoAtNet-7[*] | T-H[*] | 65k | 19.7B[PT] +32.8B | 1.5B | 85.7 | 80.6 | 95.7 | 85.6 |
| LIT [16] | ViT-g[*] | T-g | 32k | 25.8B[PT] + 18.2B | 1.1B | 84.5 | 78.7 | 93.9 | 79.4 |
| ALIGN [10] | EffNet-L2 | T-L[*] | 16k | 19.8B | $\sim$ 410M | 76.4 | 70.1 | 92.2 | 75.8 |
| CLIP [7] | ViT-L/14[†] | T-B | 32k | 12.8B | $\sim$ 200M | 76.2 | 70.1 | 88.9 | 77.2 |
| LIMoE | H/14 | | 21k | 23.3B | 675M | 84.1 | 77.7 | 94.9 | 78.7 |

## 4  Ablations

We use a smaller setup to study various aspects of LIMoE. We train B/16 models at batch size 8096 for 100,000 steps (see Appendix A.2 for further details). Table 2 shows the average over three trials of this setting alongside dense one-tower and two-tower baselines. LIMoE greatly outperforms both dense models on ImageNet 0- and 10-shot, while confidence intervals overlap for retrieval with two towers. The two-tower model is twice as large and expensive, and still falls behind the sparse one.

### 4.1  Routing and auxiliary losses

**Choice of auxiliary losses.** With the introduction of the entropy based losses in addition to classic ones, there are 7 possible auxiliary losses. We aimed to find the simplest combination of these which obtains good performance. To study this, we performed a large sweep of auxiliary losses: for $N \in [2, \dots, 5]$, we considered all $\binom{7}{N}$ possible loss combinations. Table 3 shows, for each loss, the highest performing model with and without that loss. Some conclusions stand out: Both entropy losses are important for text, but for images, the global loss is not impactful and the local

Table 2: Baselines for ablations: B/16 with batch size 8096 trained for for 100,000 steps. 0shot and 10shot columns show accuracy (%), t2i and i2t show recall@1 (%).

| Model | i1k 0shot | i1k 10shot | coco t2i | coco i2t |
|---|---|---|---|---|
| dense one-tower | $49.8\,^{50.4}_{49.2}$ | $43.8\,^{44.3}_{43.3}$ | $23.7\,^{24.0}_{23.4}$ | $36.7\,^{38.9}_{34.6}$ |
| dense two-tower | $54.7\,^{55.2}_{54.1}$ | $47.1\,^{47.6}_{46.7}$ | $26.6\,^{27.1}_{26.2}$ | $41.3\,^{42.0}_{40.6}$ |
| LIMoE | $56.9\,^{57.1}_{56.7}$ | $50.5\,^{50.8}_{50.2}$ | $25.6\,^{27.3}_{23.9}$ | $39.7\,^{42.2}_{37.1}$ |

Table 3: Across 121 combinations, each row shows the best accuracy (%) of all combinations that *included* the auxiliary loss (✓) vs. those that did not (✗). Bold auxiliary losses indicate they are in LIMoE. Validation accuracy is the average contrastive accuracy in a minibatch of size 1024.

| Auxiliary loss | Validation | | 0shot | | 10shot | |
|---|---|---|---|---|---|---|
| | ✗ | ✓ | ✗ | ✓ | ✗ | ✓ |
| Importance | 70.5 | 70.6 | 55.4 | 56.2 | 51.1 | 51.3 |
| **Load** | 70.3 | 70.6 | 56.2 | 55.7 | 51.3 | 51.1 |
| **Z-Loss** | 70.3 | 70.6 | 55.8 | 56.2 | 50.5 | 51.3 |
| **Global Ent Image** | 70.6 | 70.5 | 56.0 | 56.2 | 50.8 | 51.3 |
| **Global Ent Text** | 69.1 | 70.6 | 54.3 | 56.2 | 51.1 | 51.3 |
| Local Ent Image | 70.6 | 68.7 | 56.2 | 53.5 | 51.3 | 47.5 |
| **Local Ent Text** | 67.2 | 70.6 | 53.3 | 56.2 | 47.5 | 51.3 |

loss is harmful. The final combination of losses was chosen based on validation accuracy alongside qualitative observations around training stability and routing success rate.

**Threshold for global entropy losses.** In Section 2.2.2, we introduced a threshold $\tau$ to encourage balanced expert distributions without forcing all modalities to use all experts. To understand the importance of this threshold, we sweep over it for both the image and text global entropy losses. Appendix B.2 contains a full analysis; the most important conclusions are:

- $\tau_{\text{image}}$ did not affect the number of experts used for images, as global entropy was always high. Aside from these threshold experiments with very high $\tau_{\text{image}}$, this loss is usually inactive. It was used in our main experiments, but can likely be removed in future work.

- The threshold $\tau_{\text{text}}$ behaved exactly as a soft minimum for text experts: Sweeping $\tau_{\text{text}}$, we typically observed approximately $S = e^{\tau_{\text{text}}}$ text experts.

- Performance is robust to different values of $\tau_{\text{text}}$, provided it is not too low. A low $\tau_{\text{text}}$ can be useful to limit the number of text experts, for later pruning, see Appendix E.4.

**Mutual-information auxiliary loss.** In Section 2.2.2, we discussed an alternative loss, namely $-\text{MI}(\text{experts}; m)$, based on the mutual information between experts and modalities. While it has the advantage of merging the local and global entropy losses for both the text and image modalities into a single term, without threshold parameters, it leads to slightly worse results: in a comparable setup, it had 1.5% and 0.1% worse zero-shot and 10-shot performance compared to Table 2.

**The effect of modality balancing.** Our models use a text sequence length of 16, but image sequence lengths from 49 to 400 (for these ablations, 196).

Our ablations reveal that the entropy losses are most important when applied to the text tokens. This leads to a hypothesis that these are only necessary or useful in the imbalanced case. To test this, we vary the modality balance of LIMoE-B/16 by varying the patch size; this enables us to control the number of image tokens, and hence image:text balance, without changing the information content in the data. Fig-

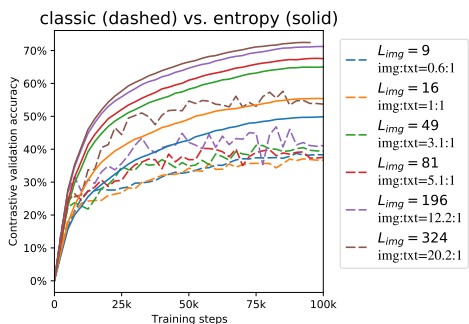

classic (dashed) vs. entropy (solid)

Figure 5: **Entropy losses are not just addressing a modality imbalance**. With different image:text balancing, including completely balanced, the entropy losses substantially improves over the classic setting.

ure 5 shows the results. First, we observe that, with entropy routing, a longer image sequence length is always better. This shows that entropy routing can effectively handle highly imbalanced setups, and mirrors the observation that for classical Vision Transformers: a longer sequence is better. Importantly, entropy routing is always far superior to the classical setup with growing gaps, even when the modalities are balanced 1:1 ($L_{\text{img}} = 16$). This experiment also confirms the robustness of entropy routing to different setups.

**Batch priority routing as a training stabilizer.** Figure 6 shows the effect of BPR during training. BPR not only ameliorates against token dropping, but also improves training stability. Models with no dispatch order intervention (first-in-first-out) perform extremely poorly, whether we route images first or text first. These routers have low success rate. Randomly shuffling tokens (i.e. deciding which tokens to drop at random when an expert becomes full) partially ameliorates this, but its performance is still much worse than that of models trained with BPR. We further analyse BPR in Appendix F.5 and show that it does not simply rank one modality above another.

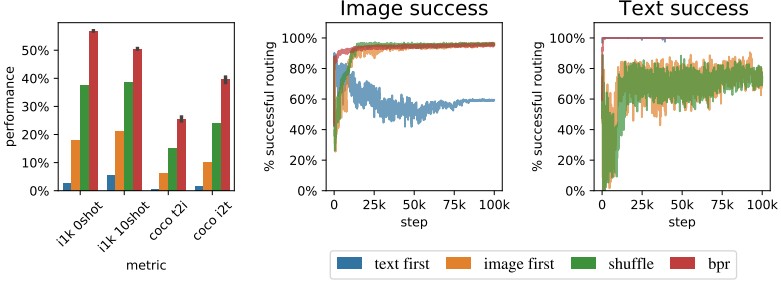

Figure 6: **BPR stabilizies training and enables performant models**; the first figure shows different performance metrics. The last two show *success rates* for the MoE router in Layer 9.

## 4.2 Other ablations

We summarize our other ablations here due to space constraints; details can be found in Appendix E.

**Router structure** (*Appendix E.3*). Our router is modality agnostic; we experiment with per-modality routers, and separate pools of per-modality experts. We find they all perform comparably to our generic, modality agnostic setup, but that separate pools of experts by design is more stable and does not require auxiliary losses for regularisation—while harder to scale to many modalities and tasks.

**Increasing selected experts per token** $K$ (*Appendix E.1*). We propose modifications to BPR and the local auxiliary loss to generalise to $K > 1$; by doing so we can steadily increase performance by increasing $K$, e.g. from 55.5% zero-shot accuracy with $K = 1$ to 61.0% with $K = 5$.

**Total number experts** (*Appendix E.2*). We show that increasing the pool of available experts at fixed $K$ improves performance (unlike what was observed for vision-only tasks [1]).

**Expert pruning** (*Appendix E.4*). We show using simple heuristics we can prune down to modality-specific experts for unimodal forward passes, thus avoiding expert collapse under unimodal batches.

**Training on public data** (*Appendix E.6*) The majority of LIMoE models were trained on proprietary data [16]. We show that LIMoE works similarly well on publically available data, retaining performance improvements against a comparable dense model.

## 5 Model Analysis

In this section, we explore some of the internal workings of LIMoE. We use simple B/32 and B/16 models with 8 experts, and the large H/14 with 32. See Appendix F for further details and experiments.

**Multimodal experts arise** (*Appendix F.1*). Aside from encouraging diversity, we do not explicitly enforce experts to specialize. Nonetheless, we observe the emergence of both modality-specific experts, and multimodal experts which process both images and texts (per-expert distributions in F.1).

**Qualitative analysis** *(Appendix F.2).* We analyse some example data and show a clear emergence of semantically meaningful experts. With images for instance, some experts specialize on lower level features (colours, lines) while others on more complex features (faces and text), see Figure 2.

**BPR ranking** *(Appendix F.5).* The local loss encourages high max-routing weights for text, and BPR ranks according to this. We show however that this does not mean text is always prioritised first: Especially in later layers, the model often prioritises important image patches over text.

# 6   Related work

Unimodal, task-specific neural networks have long been researched, with increasing convergence towards Transformer-based architectures [23, 26] for both NLP [27] and Computer Vision [13, 28, 29]. *Multimodal models* aim to process multiple types of data using a single neural network.

Many approaches "fuse" modalities [30, 31, 32, 33] to tackle inherently multimodal tasks. LIMoE is more similar to approaches which do not do that, and still operate as unimodal feature extractors. Some co-train on distinct tasks [34, 35, 36, 22] without aligning or fusing representations—effectively sharing weights across tasks—whereas others include both unimodal aspects and fused multimodal aspects for functionality in both contexts [37].

We build on deep *Sparse Mixture of Experts models*, which have been studied independently in Computer Vision [1, 2] and NLP [14, 3, 8], typically in the context of transfer learning. These models use a learned gating mechanism whereby only a subset of $K$ experts out of $E \gg K$ are activated for a given input. Many works aim to improve the gating mechanism itself, by making it differentiable [38], reformulating as a linear assignment task [39] or even swapping it out for a simple hashing algorithm [40]. MoE models have also been studied for multitask learning [38], with per-task routers [6] but a shared pool of experts. To our knowledge, sparse models have not been explored for multimodal learning.

A large body of research exists on contrastive learning, usually in self-supervised [41] but also in supervised regimes [42]. *Multimodal contrastive learning* trains on aligned data from multiple modalities. Originally studied for medical images and reports [11], it was recently scaled to noisy web data [7, 10], where strong image-text alignments enabled performant image classification and cross-modal image-text retrieval without finetuning on downstream data. Follow up works improved upon this significantly by scaling up and using pretrained models [18, 16] and multitask training with generative modelling [25] or other vision tasks [43]. These works use unimodal models which *separately* process image and text data; we are not aware of previous research using a single model to process both images and texts for contrastive learning, neither with dense nor with sparse models.

# 7   Conclusions and Future Work

We have presented LIMoE, the first multimodal sparse mixture of experts model. We uncovered new failure modes specific to this setup and proposed entropy based auxiliary losses which stabilises training and results in highly performant models. It works across many model scales, with average improvements over FLOP-matched dense baselines of +10.2% zero-shot accuracy. When scaled to a large H/14 model, we achieve 84.1% accuracy, competitive with current SOTA approaches.

**Societal impact and limitations**: The potential harms of large scale models [44], contrastive models [7] and web-scale multimodal data [45] also carry over here, as LIMoE does not explicitly address them. On the other hand, it has been shown that *pruning* models tends to cause low-resource groups to be forgotten [46], causing performance to disproportionally drop for some subgroups. This would be worth considering for our expert-pruning experiments, but by analogue, the ability to scale models with experts that can specialize deeply may result in better performance on underrepresented groups.

Environmentally speaking, training large models is costly, though efforts are made to use efficient datacenters and offset emitted $CO_2$. Prior works however show that most environmental impact occurs during model inference, and that MoEs are significantly more efficient in that regard [47]; LIMoE is naturally a good candidate for efficient, large-scale multimodal foundation models.

**Future work:** There are many interesting directions from here. The routing interference with multiple modalities still is not fully understood. In general, conclusions from applications of MoEs

to NLP have not carried over perfectly to Vision, and vice-versa, and here we see again different behaviour between images and text. Naturally, extensions to more modalities should be explored; even with only two we see fascinating interactions between different data types and the routing algorithms, and that will only get more difficult, and interesting, with more modalities.

There are always more modalities to learn, and larger models to build: sparse models provide a very natural way to scale up while juggling very different tasks and data, and we look forward to seeing more research in this area.

## 8 Acknowledgements

We first thank Andreas Steiner, Xiao Wang and Xiaohua Zhai, who led early explorations into dense single-tower models for contrastive multimodal learning, and also were instrumental in providing data access. We also thank Andreas Steiner, and Douglas Eck, for early feedback on the paper. We thank André Susano Pinto, Maxim Neumann, Barret Zoph, Liam Fedus, Wei Han and Josip Djolonga for useful discussions, and Erica Moreira and Victor Gomes for help scaling up to LIMoE-H/14.

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
