# A  Training details

All models were trained with adafactor, using the same modifications used for ViT-G [17]. Unless otherwise specified, we use learning rate $1 \times 10^{-3}$ and decoupled weight decay of magnitude $1 \times 10^{-5}$. We use a cosine learning rate decay schedule, with a linear warmup (40k steps for longer scaling study models, 10k steps for ablations). Models were trained on a mixture of Cloud TPU-v2, v3 and v4 pods.

Models were trained with 32 experts, with experts placed every 2 layers – except where explicitly stated. Otherwise, architecture parameters (e.g. hidden size, number of layers) follow those of ViT [13]. All models except for `LIMoE-H/14` use dimensionality 512 for the final output representation; this final representation is cast to bfloat16 precision for reduced all-to-all costs and increased memory efficiency. The learned contrastive temperature parameter is initialised at 10. Text sequences are tokenized to a sequence length of 16 using the T5 SentencePiece vocabulary [48]. Images were linearly renormalized to a value range of `[-1, 1]`.

## A.1  Scaling study

We train models at batch size 16,384 for 781,250 steps at resolution 224. This trains for the same number of examples as CLIP [7]; they however use a larger batch size (32768), increase resolution in the final epoch, and use a larger dimensionality for the final contrastive feature representation, all of which improve performance.

## A.2  Ablations

These are B/16 models trained for 100,000 steps at batch size 8192. The threshold used for the text global entropy loss is $\tau_T = \log(9)$ – that is, we incentivize the use of at least 9 experts (uniformly) or more (not necessarily in a uniform way). For images, $\tau_T = \log(20)$, but with this threshold, the loss is not applied at all and it can be ignored.

## A.3  `LIMoE`-H/14

The largest scale model is trained at batch size 21502, with resolution 288 and text sequence length 16. The global entropy loss thresholds are $\tau_{\text{text}} = \log(4)$ and $\tau_{\text{text}} = \log(25)$ for text and image respectively. There are MoE layers in 12 encoder blocks, namely, in 3, 7, 11, 15, 18, 21, 24, 26, 28, 30, 31, 32. The default training data is mixed with data from JFT-4B with a ratio of 3:1. Text strings are generated from JFT-4B by simply concatenating the class names. JFT-4B was also deduplicated using the same method as previous works [16].

**Checkpoint souping**. We adapt the methodology developed for finetuning [21], but instead combine checkpoints from the same run. We used a reverse-sqrt schedule [48], which has a linear cooldown at the end. To generate diversity for the model soup, we launched multiple cooldowns, and greedily selected checkpoints to maximize zero-shot accuracy on the ImageNet validation set, using the smaller subset of prompts from CLIP [7]. Checkpoints could be reused multiple times.

The model was trained for 700k steps pre-cooldown. There was one cooldown of length 125k steps from the final step, and 3 of length 40k steps starting from step 650k. Two of the cooldowns had no changes to the original setup described above. To generate diversity for the soup, we also trained one 40k cooldown with only JFT data, and one with no JFT data at all.

Figure 7 shows the zero-shot accuracy evaluated at 12.5k step intervals during training, for all the different cooldowns, and the end of training. The final model soup consisted of 8 checkpoints in total.

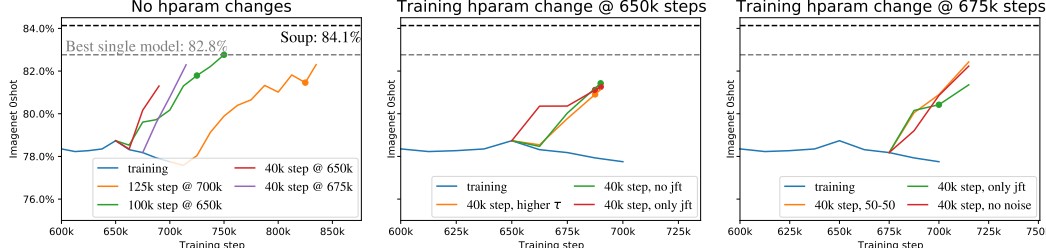

Figure 7: **Souping procedure for** `LIMoE`**-H/14**. Dots show checkpoints in the final model soup.

# B  Auxiliary losses

## B.1  Definitions of all the auxiliary losses

In Section 4.1, we study multiple combinations of auxiliary losses. For completeness, we recall below all their definitions. Given a token $\mathbf{x} \in \mathbb{R}^D$, we denote by $g(\mathbf{x}) = \texttt{softmax}(\mathbf{W}\mathbf{x}) \in \mathbb{R}^E$ the gating weights across the $E$ experts, with $\mathbf{W} \in \mathbb{R}^{E \times D}$ being the routing parameters. When we deal with a batch of multiple tokens $\{\mathbf{x}_i\}_{i=1}^n$, we use the notation $\mathbf{X} \in \mathbb{R}^{n \times D}$.

**Importance loss.** We consider the definition from [1], inspired by the original proposal of [14]. The importance loss $\Omega_{\text{imp}}$ enforces a balanced profile of the gating weights across the experts. More formally, for any expert $e \in \{1, \ldots, E\}$, we consider

$$\text{imp}_e(\mathbf{X}) = \sum_{\mathbf{x} \in \mathbf{X}} g(\mathbf{x})_e$$

and define the loss $\Omega_{\text{imp}}$ via the squared coefficient of variation for $\text{imp}(\mathbf{X}) = \{\text{imp}_e(\mathbf{X})\}_{e=1}^E$, namely

$$\Omega_{\text{imp}}(\mathbf{X}) = \left( \frac{\texttt{std}(\text{imp}(\mathbf{X}))}{\texttt{mean}(\text{imp}(\mathbf{X}))} \right)^2.$$

**Load loss.** Like previously, we follow [1] whose definition is inspired by the original proposal of [14]. We assume throughout that paragraph that the gating weights $g_{\text{noisy}}(\mathbf{x})$ are obtained by a noisy version of the routing, i.e., $g_{\text{noisy}}(\mathbf{x}) = \texttt{softmax}(\mathbf{W}\mathbf{x} + \varepsilon)$ with $\varepsilon \sim \mathcal{N}(\mathbf{0}, \sigma^2 \mathbf{I})$ and $\sigma = 1/E$ (see details in [1]). We introduce $\eta_K$ the $K$-th largest entry of $\mathbf{W}\mathbf{x} + \varepsilon$.

The load loss $\Omega_{\text{load}}$ complements the importance loss $\Omega_{\text{imp}}$ by trying to balance the *number of assignments* across the experts. To circumvent the fact that the assignments are discrete, $\Omega_{\text{imp}}$ focuses instead on the probability of selecting the expert. For any $e \in \{1, \ldots, E\}$, the probability is understood as the probability of having the expert $e$ still being among the Top-$K$ while resampling only the noise of that expert. More formally, this corresponds to

$$p_e(\mathbf{x}) = 1 - \Phi\left( \frac{\eta_K - (\mathbf{W}\mathbf{x})_e}{\sigma} \right)$$

with $\Phi$ the cumulative distribution function of a Gaussian distribution.

The load loss $\Omega_{\text{load}}$ is eventually defined by

$$\Omega_{\text{load}}(\mathbf{X}) = \left( \frac{\texttt{std}(\text{load}(\mathbf{X}))}{\texttt{mean}(\text{load}(\mathbf{X}))} \right)^2 \text{ with } \text{load}(\mathbf{X}) = \{\text{load}_e(\mathbf{X})\}_{e=1}^E \text{ and } \text{load}_e(\mathbf{X}) = \sum_{\mathbf{x} \in \mathbf{X}} p_e(\mathbf{x}).$$

**Z-loss.** The z-loss $\Omega_{\text{zloss}}$ introduced in [4] aims at controlling the maximum magnitude of the router activations $\mathbf{A} = \{\mathbf{W}\mathbf{x}_i\}_{i=1}^n \in \mathbb{R}^{n \times E}$ with entries $a_{i,e} = (\mathbf{W}\mathbf{x}_i)_e$. The loss is defined by

$$\Omega_{\text{zloss}}(\mathbf{X}) = \frac{1}{n} \sum_{i=1}^n \left( \log \left( \sum_{e=1}^E \exp(a_{i,e}) \right) \right)^2.$$

**The mutual-information loss $-\textbf{MI}(\texttt{experts}; m)$.** In Section 2.2.2, we allude to a variant of the local and global entropy losses in the form of the mutual information between the experts and the modalities (as a reminder, the sum of the local and global entropy losses corresponds instead to the (negative) mutual information between the experts and tokens, conditioned on the modality). Let us assume we have a total of $M$ modalities. Formally, and reusing the notation from Section 2.2.2, we define $-\text{MI}(\texttt{experts}; m)$ as

$$-\text{MI}(\texttt{experts}; m) = \frac{1}{M} \sum_{m'=1}^M \mathcal{H}(\tilde{p}_{m'}(\texttt{experts})) - \mathcal{H}\left( \frac{1}{M} \sum_{m'=1}^M \tilde{p}_{m'}(\texttt{experts}) \right)$$

where, for each modality $m'$, we have computed the approximate marginal probability over the $n_{m'}$ tokens of that modality

$$\tilde{p}_{m'}(\texttt{experts}) = \frac{1}{n_{m'}} \sum_{i=1}^{n_{m'}} p_{m'}(\texttt{experts}|\mathbf{x}_i)$$

and $\mathcal{H}$ denotes the entropy.

**Final aggregated auxiliary loss.** When considering the combination of several auxiliary losses, the final auxiliary loss is computed as the average over all the losses. The average is weighted by a single regularization parameter that is a hyperparameter of our approach. After some preliminary tuning phase, we have set its value to $0.04$ in all our experiments and found this choice to be robust.

## B.2 In-depth analysis of global entropy threshold

Note again that we can view a threshold $\tau$ as a soft minimum, as the minimum number of experts which must be used by a modality to satisfy the loss is $S = e^{\tau}$. We find it more intuitive to think in terms of this soft minimum threshold $S_T$.

**Performance**. Figure 8 shows the effect of the threshold on performance.

There are three phenomenon of note:

1. When the text threshold is too low, models are unstable and performance is poor.
2. Past some limit however, performance of models w.r.t. text threshold is fairly consistent.
3. Outside (and probably inside) the unstable region, the image threshold makes no systematic difference.

Model performances as a function of global entropy threshold $\tau = \log S$

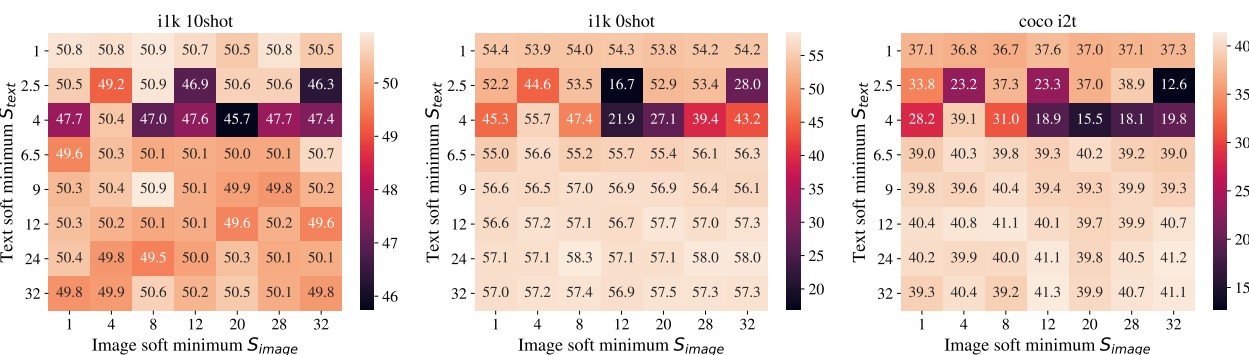

Figure 8: **A high enough text threshold encourages stability**, but otherwise performance is somewhat invariant to the thresholds used. Note the plotted quantity is the *soft minimum* $S_T = e^{\tau}$.

**Actual global entropies**. Looking at the actual entropies of model routing helps at least explain why the image threshold is unimportant. Figure 9 shows the empirical entropy. The image entropy is always large; note that when it is higher than the threshold $\tau$, the loss is not applied; ergo, for most of the settings, the global entropy loss is *not applied to images*. This also applies to almost all models trained for this paper. On the other hand, analysing text entropies, it is clear that the model closely tracks the threshold $\tau_{\text{text}}$. As a side effect, image entropy tends to reduce as $\tau_{\text{text}}$ increases.

**Expert specialization**. As discussed, the threshold can be viewed as setting an implicit soft minimum $S_T = e^{\tau}$. The number of experts actually used for each modality is shown in Figure 10. The text threshold exactly behaves as a soft minimum; as it is increased, the model has more text experts and less image experts.

**Overall**. For text, the entropy loss behaves as expected; as it is increased, there are more text experts. A few questions remain: why does it not impact performance? Why does text behave differently than images - is it due to the imbalance between them during training, or is it simply a fundamental difference in routing behavior for the two modalities?

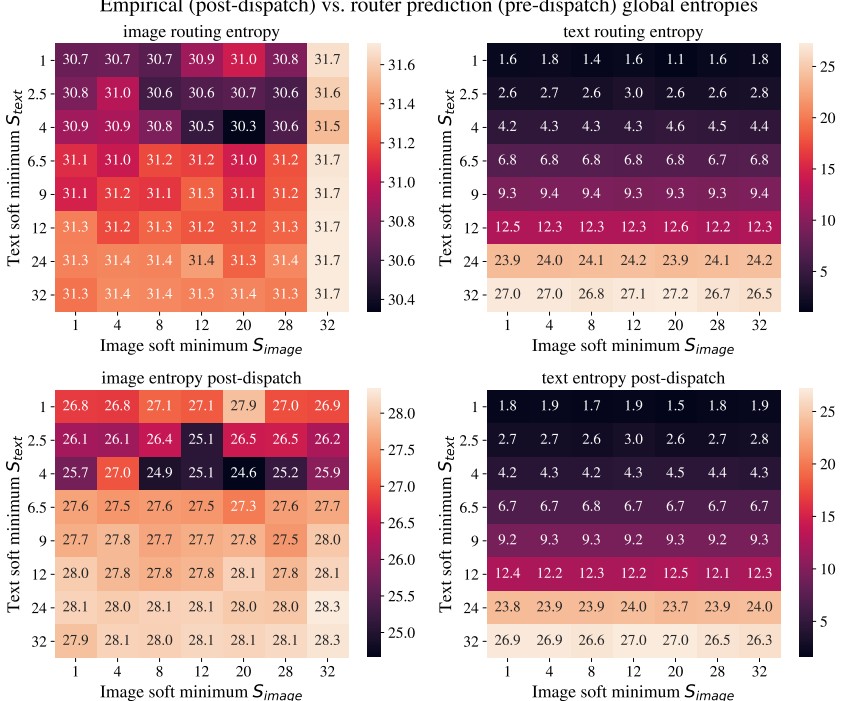

Figure 9: **Image entropy is always high, but text entropy closely tracks the target threshold**. *Top*: The routing entropy is the global entropy of the predictions of the router, which is what is actually regularised. *Bottom*: The post-dispatch entropy is the entropy of the distribution after top-K selection and capacity limits (token dropping) have interfered. For text tokens pre- and post-dispatch entropies pretty much coincide as their routing probabilities are high and BPR favors them –so little dropping happens. The story is a bit different for image tokens; some are dropped, and the pre- and post-dispatch entropies are not completely equivalent.

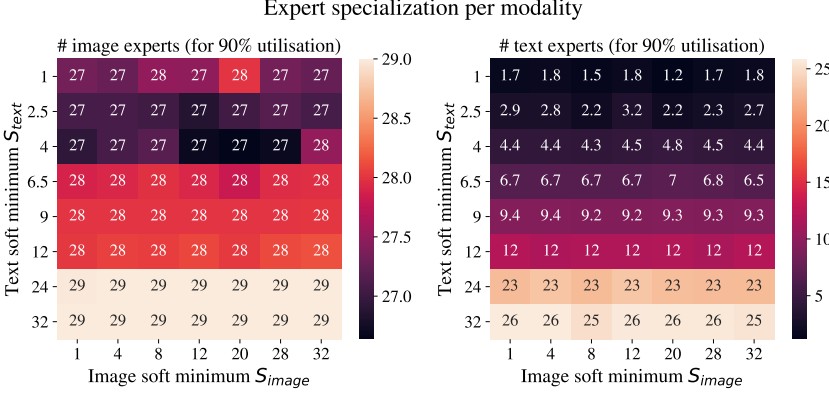

Figure 10: **Almost all experts process images, but the number of text experts closely follows the threshold**. Some experts process both modalities, so it is non trivial to classify an expert as being of a given modality. Our proxy for the number of modality specific experts is the number of experts needed to reach some routing success rate: if only 3 experts are needed for 90% text success rate, then text routing is not highly distributed.

# C  Tabular results

## C.1  Scaling comparison, and architecture definitions.

All results and parameters from Figure 4 are shown in Table 4, alongside the results of `LIMoE-H/14`.

Table 4: The results from Figure 4 and `LIMoE-H/14`.

| Model | Patch size | Layers | Heads | Hidden size | MLP size | i1k 0shot | i1k 10shot | coco t2i | coco i2t |
|---|---|---|---|---|---|---|---|---|---|
| Dense S/32 | 32 | 8 | 8 | 512 | 2048 | 44.4 | 33.8 | 18.2 | 30.4 |
| LIMoE S/32 | | | | | | 57.9 | 48.1 | 25.0 | 38.9 |
| Dense S/16 | 16 | 8 | 8 | 512 | 2048 | 50.3 | 40.4 | 22.7 | 35.8 |
| LIMoE S/16 | | | | | | 64.5 | 56.3 | 29.2 | 43.7 |
| Dense B/32 | 32 | 12 | 12 | 768 | 3072 | 58.8 | 48.7 | 27.4 | 42.5 |
| LIMoE B/32 | | | | | | 67.5 | 60.4 | 31.0 | 45.7 |
| Dense B/16 | 16 | 12 | 12 | 768 | 3072 | 64.3 | 55.3 | 31.7 | 46.8 |
| LIMoE B/16 | | | | | | 73.7 | 68.2 | 36.2 | 51.3 |
| Dense L/32 | 32 | 24 | 16 | 1024 | 4096 | 68.1 | 60.7 | 34.6 | 51.2 |
| LIMoE L/32 | | | | | | 74.6 | 69.7 | 37.2 | 54.5 |
| Dense L/16 | 16 | 24 | 16 | 1024 | 4096 | 73.5 | 67.6 | 38.3 | 54.3 |
| LIMoE L/16 | | | | | | 78.6 | 74.7 | 39.6 | 55.7 |
| LIMoE-H/14 | 14 | 32 | 16 | 1280 | 5120 | 84.1 | 81.4 | 39.8 | 51.1 |

## C.2 All tabular results

Table 5: All models trained.

| model type | arch | notes | $\tau_{\text{text}}$ | $\tau_{\text{image}}$ | batch train | batch eval | data seen | 0shot | 10shot | coco i2t | coco t2i | val acc |
|---|---|---|---|---|---|---|---|---|---|---|---|---|
| *Figure 4:* Sweep over scale with CLIP-esque training regime | | | | | | | | | | | | |
| dense | S/32 | | - | - | 16384 | 1024 | 12.8B | 44.4 | 33.8 | 30.4 | 18.2 | 62.1 |
| LIMoE | S/32 | | log(12) | log(17) | 16384 | 1024 | 12.8B | 57.9 | 48.1 | 38.9 | 25.0 | 73.1 |
| dense | S/16 | | - | - | 16384 | 1024 | 12.8B | 50.3 | 40.4 | 35.8 | 22.7 | 67.6 |
| LIMoE | S/16 | | log(9) | log(20) | 16384 | 1024 | 12.8B | 64.5 | 56.3 | 43.7 | 29.2 | 77.1 |
| dense | B/32 | | - | - | 16384 | 1024 | 12.8B | 58.8 | 48.7 | 42.5 | 27.4 | 72.5 |
| LIMoE | B/32 | | log(12) | log(17) | 16384 | 1024 | 12.8B | 67.5 | 60.4 | 45.7 | 31.0 | 79.2 |
| dense | B/16 | | - | - | 16384 | 1024 | 12.8B | 64.3 | 55.3 | 46.8 | 31.7 | 76.4 |
| LIMoE | B/16 | | log(9) | log(20) | 16384 | 1024 | 12.8B | 73.7 | 68.2 | 51.3 | 36.2 | 82.3 |
| dense | L/32 | | - | - | 16384 | 1024 | 12.8B | 68.1 | 60.7 | 51.2 | 34.6 | 78.5 |
| LIMoE | L/32 | | log(20) | log(1) | 16384 | 1024 | 12.8B | 74.6 | 69.7 | 54.5 | 37.2 | 83.3 |
| dense | L/16 | | - | - | 16384 | 1024 | 12.8B | 73.5 | 67.6 | 54.3 | 38.3 | 82.2 |
| LIMoE | L/16 | | log(28) | log(8) | 16384 | 1024 | 12.8B | 78.6 | 74.7 | 55.7 | 39.6 | 85.9 |
| *Table 2:* The baselines for many of the ablation experiments below (1T = 1 Tower, 2T = 2 Towers) | | | | | | | | | | | | |
| dense (1T) | B/16 | Trial 0 | - | - | 8192 | 1024 | 819.2M | 49.9 | 43.7 | 37.7 | 23.7 | 66.0 |
| dense (1T) | B/16 | Trial 1 | - | - | 8192 | 1024 | 819.2M | 50.0 | 44.0 | 36.6 | 23.8 | 66.0 |
| dense (1T) | B/16 | Trial 2 | - | - | 8192 | 1024 | 819.2M | 49.5 | 43.6 | 36.0 | 23.6 | 66.0 |
| dense (2T) | B/16 | Trial 0 | - | - | 8192 | 1024 | 819.2M | 54.8 | 47.3 | 41.3 | 26.6 | 69.7 |
| dense (2T) | B/16 | Trial 1 | - | - | 8192 | 1024 | 819.2M | 54.4 | 47.0 | 41.0 | 26.5 | 69.4 |
| dense (2T) | B/16 | Trial 2 | - | - | 8192 | 1024 | 819.2M | 54.9 | 47.1 | 41.6 | 26.9 | 69.5 |
| LIMoE | B/16 | Trial 0 | log(9) | log(20) | 8192 | 1024 | 819.2M | 56.8 | 50.5 | 40.1 | 25.7 | 70.8 |
| LIMoE | B/16 | Trial 1 | log(9) | log(20) | 8192 | 1024 | 819.2M | 57.0 | 50.4 | 40.4 | 26.2 | 70.8 |
| LIMoE | B/16 | Trial 2 | log(9) | log(20) | 8192 | 1024 | 819.2M | 56.9 | 50.6 | 38.5 | 24.9 | 70.7 |
| *Table 6:* Increasing the number of selected experts, with adjustments to local entropy loss and BPR | | | | | | | | | | | | |
| LIMoE | B/16 | k=2, target entropy loss, max BPR | log(4) | log(25) | 8192 | 1024 | 819.2M | 55.9 | 48.1 | 36.6 | 25.6 | 69.1 |
| LIMoE | B/16 | k=3, target entropy loss, max BPR | log(4) | log(25) | 8192 | 1024 | 819.2M | 48.2 | 48.9 | 27.7 | 21.1 | 64.3 |
| LIMoE | B/16 | k=5, target entropy loss, max BPR | log(4) | log(25) | 8192 | 1024 | 819.2M | 11.7 | 36.4 | 7.1 | 5.6 | 23.2 |
| LIMoE | B/16 | k=2, merged loss, max BPR | log(4) | log(25) | 8192 | 1024 | 819.2M | 46.4 | 49.4 | 28.1 | 10.7 | 57.3 |
| LIMoE | B/16 | k=3, merged loss, max BPR | log(4) | log(25) | 8192 | 1024 | 819.2M | 52.6 | 47.9 | 33.0 | 23.2 | 65.5 |
| LIMoE | B/16 | k=5, merged loss, max BPR | log(4) | log(25) | 8192 | 1024 | 819.2M | 60.3 | 53.4 | 43.3 | 28.0 | 73.3 |
| LIMoE | B/16 | k=2, top1 loss, max BPR | log(4) | log(25) | 8192 | 1024 | 819.2M | 58.3 | 51.9 | 42.0 | 27.2 | 71.7 |
| LIMoE | B/16 | k=3, top1 loss, max BPR | log(4) | log(25) | 8192 | 1024 | 819.2M | 59.0 | 53.6 | 42.7 | 28.1 | 72.1 |
| LIMoE | B/16 | k=5, top1 loss, max BPR | log(4) | log(25) | 8192 | 1024 | 819.2M | 59.8 | 54.6 | 43.0 | 27.8 | 72.5 |
| LIMoE | B/16 | k=2, none loss, max BPR | log(4) | log(25) | 8192 | 1024 | 819.2M | 46.8 | 44.3 | 28.9 | 14.3 | 61.0 |
| LIMoE | B/16 | k=3, none loss, max BPR | log(4) | log(25) | 8192 | 1024 | 819.2M | 44.6 | 42.2 | 27.3 | 17.5 | 57.5 |
| LIMoE | B/16 | k=5, none loss, max BPR | log(4) | log(25) | 8192 | 1024 | 819.2M | 17.5 | 35.4 | 6.1 | 5.9 | 22.8 |
| LIMoE | B/16 | k=2, target entropy loss, sum BPR | log(4) | log(25) | 8192 | 1024 | 819.2M | 58.2 | 51.8 | 42.2 | 27.7 | 71.6 |
| LIMoE | B/16 | k=3, target entropy loss, sum BPR | log(4) | log(25) | 8192 | 1024 | 819.2M | 59.1 | 53.2 | 42.3 | 27.5 | 72.5 |
| LIMoE | B/16 | k=5, target entropy loss, sum BPR | log(4) | log(25) | 8192 | 1024 | 819.2M | 60.4 | 53.8 | 42.1 | 28.0 | 73.0 |
| LIMoE | B/16 | k=2, merged loss, sum BPR | log(4) | log(25) | 8192 | 1024 | 819.2M | 59.0 | 52.4 | 41.1 | 27.1 | 72.2 |
| LIMoE | B/16 | k=3, merged loss, sum BPR | log(4) | log(25) | 8192 | 1024 | 819.2M | 60.0 | 52.8 | 42.4 | 27.6 | 73.0 |
| LIMoE | B/16 | k=5, merged loss, sum BPR | log(4) | log(25) | 8192 | 1024 | 819.2M | 61.0 | 53.6 | 42.7 | 28.4 | 73.4 |
| *Figure 12:* Increasing the total number of available experts with fixed $k = 1$ | | | | | | | | | | | | |
| LIMoE | B/16 | Total # experts = 4 | log(2.4) | log(0.8) | 8192 | 1024 | 819.2M | 52.3 | 46.9 | 37.8 | 24.4 | 67.9 |
| LIMoE | B/16 | Total # experts = 8 | log(4.8) | log(1.6) | 8192 | 1024 | 819.2M | 54.4 | 48.2 | 39.5 | 25.5 | 69.4 |
| LIMoE | B/16 | Total # experts = 16 | log(9.6) | log(3.2) | 8192 | 1024 | 819.2M | 55.7 | 49.5 | 38.9 | 25.5 | 70.2 |
| LIMoE | B/16 | Total # experts = 32 | log(19.2) | log(6.4) | 8192 | 1024 | 819.2M | 57.3 | 50.4 | 40.1 | 26.0 | 70.9 |
| LIMoE | B/16 | Total # experts = 64 | log(38.4) | log(12.8) | 8192 | 1024 | 819.2M | 58.0 | 50.7 | 41.2 | 26.5 | 71.3 |
| *Figure 16:* Varying the group size, trading off compute efficiency and stability | | | | | | | | | | | | |
| LIMoE | B/16 | Num groups = 1 | log(9) | log(20) | 8192 | 1024 | 819.2M | 56.7 | 49.4 | 40.3 | 25.8 | 70.7 |
| LIMoE | B/16 | Num groups = 2 | log(9) | log(20) | 8192 | 1024 | 819.2M | 56.6 | 50.4 | 40.4 | 25.4 | 70.8 |
| LIMoE | B/16 | Num groups = 4 | log(9) | log(20) | 8192 | 1024 | 819.2M | 56.1 | 49.5 | 39.1 | 24.8 | 69.8 |
| LIMoE | B/16 | Num groups = 8 | log(9) | log(20) | 8192 | 1024 | 819.2M | 47.4 | 44.0 | 29.5 | 19.6 | 62.4 |
| LIMoE | B/16 | Num groups = 16 | log(9) | log(20) | 8192 | 1024 | 819.2M | 50.1 | 45.3 | 33.4 | 21.1 | 65.2 |
| LIMoE | B/16 | Num groups = 32 | log(9) | log(20) | 8192 | 1024 | 819.2M | 23.8 | 31.7 | 11.6 | 8.3 | 38.9 |
| LIMoE | B/16 | Num groups = 64 | log(9) | log(20) | 8192 | 1024 | 819.2M | 1.6 | 17.8 | 1.7 | 0.9 | 6.2 |
| LIMoE | B/16 | Num groups = 128 | log(9) | log(20) | 8192 | 1024 | 819.2M | 0.1 | 38.3 | 0.0 | 0.0 | 0.1 |
| *Figure 6:* Study different alternatives for routing dispatch ordering | | | | | | | | | | | | |
| LIMoE | B/16 | Dispatch = shuffle | log(9) | log(20) | 8192 | 1024 | 819.2M | 37.6 | 38.5 | 24.2 | 15.1 | 56.5 |
| LIMoE | B/16 | Dispatch = image first | log(9) | log(20) | 8192 | 1024 | 819.2M | 17.8 | 21.3 | 10.1 | 6.3 | 42.3 |
| LIMoE | B/16 | Dispatch = bpr | log(9) | log(20) | 8192 | 1024 | 819.2M | 56.8 | 50.5 | 40.1 | 25.7 | 70.8 |
| LIMoE | B/16 | Dispatch = bpr | log(9) | log(20) | 8192 | 1024 | 819.2M | 57.0 | 50.4 | 40.4 | 26.2 | 70.8 |
| LIMoE | B/16 | Dispatch = bpr | log(9) | log(20) | 8192 | 1024 | 819.2M | 56.9 | 50.6 | 38.5 | 24.9 | 70.7 |
| LIMoE | B/16 | Dispatch = text first | log(4) | log(25) | 8192 | 1024 | 819.2M | 0.1 | 1.6 | 0.0 | 0.0 | 3.3 |
| *Table 7:* Variations on the joint, modality agnostic router used for LIMoE | | | | | | | | | | | | |
| LIMoE | B/16 | Router = per modality | log(4) | log(25) | 8192 | 1024 | 819.2M | 56.8 | 50.5 | 40.1 | 25.6 | 70.4 |
| LIMoE | B/16 | Router = partitioned | - | - | 8192 | 1024 | 819.2M | 56.8 | 50.1 | 39.1 | 25.1 | 70.8 |
| *Figure 5:* With fixed text seq len 16, vary image seq len to study effect of modality balancing. | | | | | | | | | | | | |
| LIMoE | B/12 | Image seq len 324. Losses: classic | - | - | 8192 | 1024 | 819.2M | 40.8 | 42.5 | 23.4 | 15.6 | 53.7 |
| LIMoE | B/16 | Image seq len 196. Losses: classic | - | - | 8192 | 1024 | 819.2M | 17.5 | 32.1 | 11.9 | 8.9 | 41.1 |
| LIMoE | B/24 | Image seq len 81. Losses: classic | - | - | 8192 | 1024 | 819.2M | 28.9 | 33.4 | 12.3 | 10.5 | 37.5 |
| LIMoE | B/32 | Image seq len 49. Losses: classic | - | - | 8192 | 1024 | 819.2M | 29.9 | 29.8 | 12.0 | 8.8 | 39.4 |
| LIMoE | B/48 | Image seq len 16. Losses: classic | - | - | 8192 | 1024 | 819.2M | 26.8 | 24.2 | 13.0 | 8.2 | 36.7 |

Table 5: All models trained.

| model type | arch | notes | $\tau_{\text{text}}$ | $\tau_{\text{image}}$ | batch train | batch eval | data seen | 0shot | 10shot | coco i2t | coco t2i | val acc |
|---|---|---|---|---|---|---|---|---|---|---|---|---|
| LIMoE | B/64 | Image seq len 9. Losses: classic | - | - | 8192 | 1024 | 819.2M | 24.8 | 21.2 | 11.7 | 7.7 | 38.3 |
| LIMoE | B/12 | Image seq len 324. Losses: entropy | $\log(25)$ | $\log(1)$ | 8192 | 1024 | 819.2M | 58.1 | 50.4 | 40.4 | 27.0 | 72.4 |
| LIMoE | B/16 | Image seq len 196. Losses: entropy | $\log(25)$ | $\log(1)$ | 8192 | 1024 | 819.2M | 57.2 | 50.3 | 40.4 | 26.4 | 71.2 |
| LIMoE | B/24 | Image seq len 81. Losses: entropy | $\log(25)$ | $\log(1)$ | 8192 | 1024 | 819.2M | 54.1 | 45.4 | 37.3 | 24.0 | 67.5 |
| LIMoE | B/32 | Image seq len 49. Losses: entropy | $\log(25)$ | $\log(1)$ | 8192 | 1024 | 819.2M | 50.5 | 41.4 | 35.2 | 21.4 | 65.0 |
| LIMoE | B/48 | Image seq len 16. Losses: entropy | $\log(25)$ | $\log(1)$ | 8192 | 1024 | 819.2M | 40.7 | 31.0 | 26.4 | 15.2 | 55.4 |
| LIMoE | B/64 | Image seq len 9. Losses: entropy | $\log(25)$ | $\log(1)$ | 8192 | 1024 | 819.2M | 31.8 | 24.6 | 20.6 | 10.9 | 49.9 |
| *Appendix B.2:* Varying global entropy thresholds $\tau_{\text{text}}$ and $\tau_{\text{image}}$ independently | | | | | | | | | | | | |
| LIMoE | B/16 | | $\log(1)$ | $\log(1)$ | 8192 | 1024 | 819.2M | 54.4 | 50.8 | 37.1 | 24.4 | 69.1 |
| LIMoE | B/16 | | $\log(1)$ | $\log(4)$ | 8192 | 1024 | 819.2M | 53.9 | 50.8 | 36.8 | 24.1 | 68.9 |
| LIMoE | B/16 | | $\log(1)$ | $\log(8)$ | 8192 | 1024 | 819.2M | 54.0 | 50.9 | 36.7 | 24.1 | 68.8 |
| LIMoE | B/16 | | $\log(1)$ | $\log(12)$ | 8192 | 1024 | 819.2M | 54.3 | 50.7 | 37.6 | 23.8 | 68.6 |
| LIMoE | B/16 | | $\log(1)$ | $\log(20)$ | 8192 | 1024 | 819.2M | 53.8 | 50.5 | 37.0 | 24.1 | 68.5 |
| LIMoE | B/16 | | $\log(1)$ | $\log(28)$ | 8192 | 1024 | 819.2M | 54.2 | 50.8 | 37.1 | 24.1 | 69.1 |
| LIMoE | B/16 | | $\log(1)$ | $\log(32)$ | 8192 | 1024 | 819.2M | 54.2 | 50.5 | 37.3 | 24.2 | 69.1 |
| LIMoE | B/16 | | $\log(2.5)$ | $\log(1)$ | 8192 | 1024 | 819.2M | 52.2 | 50.5 | 33.8 | 19.6 | 67.0 |
| LIMoE | B/16 | | $\log(2.5)$ | $\log(4)$ | 8192 | 1024 | 819.2M | 44.6 | 49.2 | 23.2 | 17.3 | 57.7 |
| LIMoE | B/16 | | $\log(2.5)$ | $\log(8)$ | 8192 | 1024 | 819.2M | 53.5 | 50.9 | 37.3 | 23.8 | 69.6 |
| LIMoE | B/16 | | $\log(2.5)$ | $\log(12)$ | 8192 | 1024 | 819.2M | 16.7 | 46.9 | 23.3 | 14.1 | 49.4 |
| LIMoE | B/16 | | $\log(2.5)$ | $\log(20)$ | 8192 | 1024 | 819.2M | 52.9 | 50.6 | 37.0 | 23.9 | 69.6 |
| LIMoE | B/16 | | $\log(2.5)$ | $\log(28)$ | 8192 | 1024 | 819.2M | 53.4 | 50.6 | 38.9 | 24.5 | 69.7 |
| LIMoE | B/16 | | $\log(2.5)$ | $\log(32)$ | 8192 | 1024 | 819.2M | 28.0 | 46.3 | 12.6 | 8.2 | 48.5 |
| LIMoE | B/16 | | $\log(4)$ | $\log(1)$ | 8192 | 1024 | 819.2M | 45.3 | 47.7 | 28.2 | 20.8 | 64.2 |
| LIMoE | B/16 | | $\log(4)$ | $\log(4)$ | 8192 | 1024 | 819.2M | 55.7 | 50.4 | 39.1 | 25.0 | 70.3 |
| LIMoE | B/16 | | $\log(4)$ | $\log(8)$ | 8192 | 1024 | 819.2M | 47.4 | 47.0 | 31.0 | 18.2 | 63.9 |
| LIMoE | B/16 | | $\log(4)$ | $\log(12)$ | 8192 | 1024 | 819.2M | 21.9 | 47.6 | 18.9 | 18.8 | 55.1 |
| LIMoE | B/16 | | $\log(4)$ | $\log(20)$ | 8192 | 1024 | 819.2M | 27.1 | 45.7 | 15.5 | 17.0 | 51.8 |
| LIMoE | B/16 | | $\log(4)$ | $\log(28)$ | 8192 | 1024 | 819.2M | 39.4 | 47.7 | 18.1 | 14.4 | 49.4 |
| LIMoE | B/16 | | $\log(4)$ | $\log(32)$ | 8192 | 1024 | 819.2M | 43.2 | 47.4 | 19.8 | 15.5 | 54.2 |
| LIMoE | B/16 | | $\log(6.5)$ | $\log(1)$ | 8192 | 1024 | 819.2M | 55.0 | 49.6 | 39.0 | 25.4 | 70.6 |
| LIMoE | B/16 | | $\log(6.5)$ | $\log(4)$ | 8192 | 1024 | 819.2M | 56.6 | 50.3 | 40.3 | 25.1 | 70.7 |
| LIMoE | B/16 | | $\log(6.5)$ | $\log(8)$ | 8192 | 1024 | 819.2M | 55.2 | 50.1 | 39.8 | 25.6 | 70.7 |
| LIMoE | B/16 | | $\log(6.5)$ | $\log(12)$ | 8192 | 1024 | 819.2M | 55.7 | 50.1 | 39.3 | 25.2 | 70.8 |
| LIMoE | B/16 | | $\log(6.5)$ | $\log(20)$ | 8192 | 1024 | 819.2M | 55.4 | 50.0 | 40.2 | 25.2 | 70.7 |
| LIMoE | B/16 | | $\log(6.5)$ | $\log(28)$ | 8192 | 1024 | 819.2M | 56.1 | 50.1 | 39.2 | 25.4 | 70.6 |
| LIMoE | B/16 | | $\log(6.5)$ | $\log(32)$ | 8192 | 1024 | 819.2M | 56.3 | 50.7 | 39.0 | 25.3 | 70.7 |
| LIMoE | B/16 | | $\log(9)$ | $\log(1)$ | 8192 | 1024 | 819.2M | 56.6 | 50.3 | 39.8 | 25.6 | 71.0 |
| LIMoE | B/16 | | $\log(9)$ | $\log(4)$ | 8192 | 1024 | 819.2M | 56.5 | 50.4 | 39.6 | 25.7 | 70.7 |
| LIMoE | B/16 | | $\log(9)$ | $\log(8)$ | 8192 | 1024 | 819.2M | 57.0 | 50.9 | 40.4 | 26.4 | 70.8 |
| LIMoE | B/16 | | $\log(9)$ | $\log(12)$ | 8192 | 1024 | 819.2M | 56.9 | 50.1 | 39.4 | 26.0 | 70.9 |
| LIMoE | B/16 | | $\log(9)$ | $\log(20)$ | 8192 | 1024 | 819.2M | 56.9 | 49.9 | 39.3 | 25.6 | 70.7 |
| LIMoE | B/16 | | $\log(9)$ | $\log(28)$ | 8192 | 1024 | 819.2M | 56.4 | 49.8 | 39.9 | 25.9 | 70.7 |
| LIMoE | B/16 | | $\log(9)$ | $\log(32)$ | 8192 | 1024 | 819.2M | 56.1 | 50.2 | 39.3 | 24.8 | 70.8 |
| LIMoE | B/16 | | $\log(12)$ | $\log(1)$ | 8192 | 1024 | 819.2M | 56.6 | 50.3 | 40.4 | 26.2 | 71.0 |
| LIMoE | B/16 | | $\log(12)$ | $\log(4)$ | 8192 | 1024 | 819.2M | 57.2 | 50.2 | 40.8 | 25.8 | 71.1 |
| LIMoE | B/16 | | $\log(12)$ | $\log(8)$ | 8192 | 1024 | 819.2M | 57.1 | 50.1 | 41.1 | 25.5 | 71.2 |
| LIMoE | B/16 | | $\log(12)$ | $\log(12)$ | 8192 | 1024 | 819.2M | 56.7 | 50.1 | 40.1 | 25.2 | 71.2 |
| LIMoE | B/16 | | $\log(12)$ | $\log(20)$ | 8192 | 1024 | 819.2M | 57.7 | 49.6 | 39.7 | 25.3 | 70.8 |
| LIMoE | B/16 | | $\log(12)$ | $\log(28)$ | 8192 | 1024 | 819.2M | 57.0 | 50.2 | 39.9 | 26.0 | 71.1 |
| LIMoE | B/16 | | $\log(12)$ | $\log(32)$ | 8192 | 1024 | 819.2M | 57.3 | 49.6 | 40.7 | 25.9 | 70.8 |
| LIMoE | B/16 | | $\log(24)$ | $\log(1)$ | 8192 | 1024 | 819.2M | 57.1 | 50.4 | 40.2 | 25.8 | 71.3 |
| LIMoE | B/16 | | $\log(24)$ | $\log(4)$ | 8192 | 1024 | 819.2M | 57.1 | 49.8 | 39.9 | 25.8 | 70.9 |
| LIMoE | B/16 | | $\log(24)$ | $\log(8)$ | 8192 | 1024 | 819.2M | 58.3 | 49.5 | 40.0 | 26.1 | 71.1 |
| LIMoE | B/16 | | $\log(24)$ | $\log(12)$ | 8192 | 1024 | 819.2M | 57.1 | 50.0 | 41.1 | 25.9 | 70.9 |
| LIMoE | B/16 | | $\log(24)$ | $\log(20)$ | 8192 | 1024 | 819.2M | 57.1 | 50.3 | 39.8 | 25.4 | 71.1 |
| LIMoE | B/16 | | $\log(24)$ | $\log(28)$ | 8192 | 1024 | 819.2M | 58.0 | 50.1 | 40.5 | 26.0 | 71.2 |
| LIMoE | B/16 | | $\log(24)$ | $\log(32)$ | 8192 | 1024 | 819.2M | 58.0 | 50.1 | 41.2 | 25.7 | 71.0 |
| LIMoE | B/16 | | $\log(32)$ | $\log(1)$ | 8192 | 1024 | 819.2M | 57.0 | 49.8 | 39.3 | 25.3 | 71.2 |
| LIMoE | B/16 | | $\log(32)$ | $\log(4)$ | 8192 | 1024 | 819.2M | 57.2 | 49.9 | 40.4 | 25.9 | 71.1 |
| LIMoE | B/16 | | $\log(32)$ | $\log(8)$ | 8192 | 1024 | 819.2M | 57.4 | 50.6 | 39.2 | 25.8 | 71.1 |
| LIMoE | B/16 | | $\log(32)$ | $\log(12)$ | 8192 | 1024 | 819.2M | 56.9 | 50.2 | 41.3 | 26.4 | 71.0 |
| LIMoE | B/16 | | $\log(32)$ | $\log(20)$ | 8192 | 1024 | 819.2M | 57.5 | 50.5 | 39.9 | 26.0 | 71.1 |
| LIMoE | B/16 | | $\log(32)$ | $\log(28)$ | 8192 | 1024 | 819.2M | 57.3 | 50.1 | 40.7 | 26.0 | 71.0 |
| LIMoE | B/16 | | $\log(32)$ | $\log(32)$ | 8192 | 1024 | 819.2M | 57.3 | 49.8 | 41.1 | 26.3 | 71.0 |
| *Table 8:* Training on publically available LAION400M data. | | | | | | | | | | | | |
| dense | B/16 | Trial 0 | - | - | 16384 | 1024 | 1.4B | 56.1 | 47.9 | 43.0 | 27.9 | 96.6 |
| dense | B/16 | Trial 1 | - | - | 16384 | 1024 | 1.4B | 56.0 | 47.7 | 42.8 | 27.6 | 96.5 |
| dense | B/16 | Trial 2 | - | - | 16384 | 1024 | 1.4B | 55.8 | 47.5 | 42.5 | 27.9 | 96.6 |
| LIMoE | B/16 | Trial 0 | $\log(9)$ | $\log(20)$ | 16384 | 1024 | 1.4B | 61.1 | 54.4 | 44.1 | 28.9 | 97.9 |
| LIMoE | B/16 | Trial 1 | $\log(9)$ | $\log(20)$ | 16384 | 1024 | 1.4B | 60.9 | 54.4 | 43.5 | 28.7 | 97.9 |
| LIMoE | B/16 | Trial 2 | $\log(9)$ | $\log(20)$ | 16384 | 1024 | 1.4B | 61.1 | 54.1 | 43.6 | 29.0 | 97.9 |

# D  Computational costs of `LIMoE`

## D.1  Unimodal evaluation with multimodal experts

Recall that each expert has a capacity $C$ - it can process at most $C$ tokens, and if it is assigned more, those above $C$ will not be processed. This capacity is usually set relative to some 'ideal'. If there are $N$ tokens and $E$ experts, we usually assume each expert can handle at most $C_R \times \frac{N}{E}$ tokens, where $C_R \geq 1$ is a slack factor. This way we try to reach a balanced setup where most expert process a similar number of tokens.

Multimodal routing presents a unique issue here. During training, the model learns to balance tokens when it has both images *and* text available to it. When there is only one modality, it will not use all the experts due to natural emergence of modality-specific experts - but the expert capacity size will be set *assuming all experts are used*. This results in high rates of token dropping, depending on the ratio of modality-specific experts.

In this effort, we encounter this during zero-shot classification and retrieval; models first compute representations for all text tokens, and then separately for all image tokens. In order to get around this token dropping, we simply evaluate with a high slack factor $C_R = 16$.

There are however other natural solutions; for many circumstances, one could trivially restructure evaluation such that image and text inputs are processed simultaneously. A more interesting, MoE specific solution is *pruning* modality specific experts, which is explored and shown to work in E.4. `LIMoE` models could have been evaluated at a 'normal' capacity, with pruned experts.

## D.2  Understanding the compute costs of `LIMoE`

Zero-shot evaluation on ImageNet with 6 prompts requires 6000 text forward passes and 50000 image forward passes. With 80 prompts, a la CLIP [7], it is 80000 for text instead. How does one compare compute cost vs. performance? The costs of `LIMoE`, its dense baselines, and other two-tower models, were computed assuming a full batch of images and texts, as this is the approach which makes the least assumptions about the downstream setup. This does not generalise perfectly: if, for example, a particular use case processed very large numbers of texts but only few images, models with smaller/cheaper text towers would be clearly advantaged.

**Full profiling data for Section 3**. For training and evaluation, we used a variety of TPU versions. For consistency, we profiled computation times on a TPUv3 (v3-32 to be more precise[2]). Figure 11 shows performance with respect to different proxies for compute cost. As discussed in Section 3, `LIMoE` is clearly pareto optimal with respect to total FLOPs. However, this does not fully account for certain costs related to MoE models, such as cross-device communication. Figures 11b and 11d show the performance with respect to step time. With respect to zeroshot and 10-shot classification accuracy, the performance improvements of `LIMoE` are significant enough that it is still clearly pareto optimal; for retrieval metrics on COCO, `LIMoE`'s gains exactly justify the costs, and it is not significantly more efficient than dense baselines. The story is similar whether looking at train or evaluation cost.

---

[2]`https://cloud.google.com/tpu/docs/types-topologies`

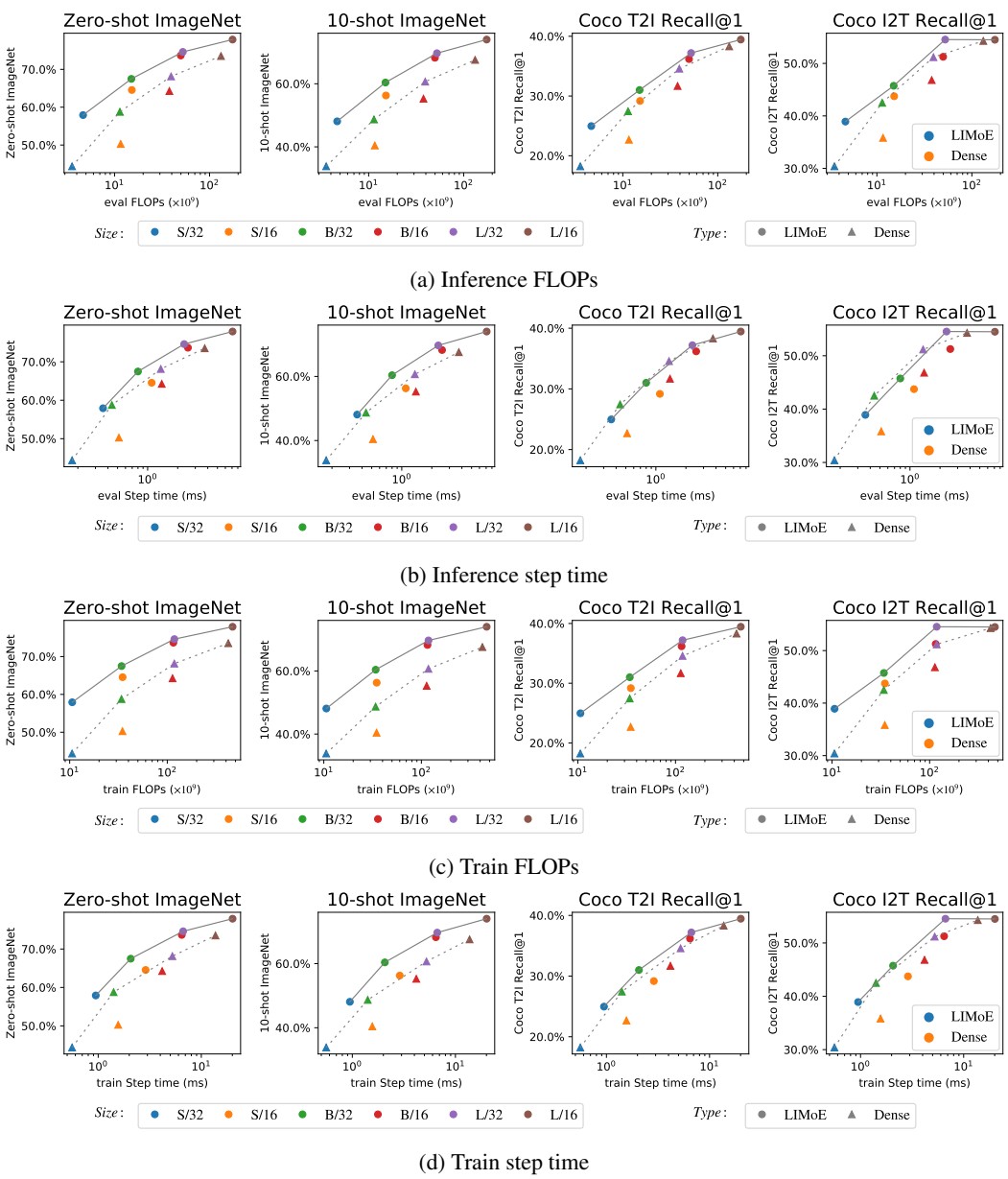

Figure 11: Pareto frontiers with respect to different measures of computational cost

Table 6: Increasing number of selected experts improves performance with appropriate changes to auxiliary losses and BPR. Table entries are ImageNet zero-shot accuracy in %.

| Local entropy method | BPR score = max | | | | BPR score = sum | |
|---|---|---|---|---|---|---|
| | None | Default | Target Ent | Merged | Target Ent | Merged |
| $K = 1$ | | $55.5^{57.7}_{53.3}$ | | | | |
| $K = 2$ | 46.8 | 58.3 | 55.9 | 46.4 | 58.2 | 59.0 |
| $K = 3$ | 44.6 | 59.0 | 48.2 | 52.6 | 59.1 | 60.0 |
| $K = 5$ | 17.5 | 59.8 | 11.7 | 60.3 | 60.4 | 61.0 |

## E Further experiments

In this section, we present further ablations not included in the main text due to space constraints.

### E.1 Increasing the number of selected experts

All models in this paper select $K = 1$ expert per token to match the cost of a dense backbone.

There are two main challenges with increasing $K$:

*Modifications to auxiliary losses.* The local entropy loss effectively encourages that router choices are one-hot. When increasing $K$, the model is still incentivized to only use 1 expert, assigning other experts weights near 0, thereby effectively behaving as $K = 1$.

We try two modifications to the local loss to ameliorate this:

- *Target entropy*: Encourage the local entropy to be $\log K$ – at least a uniform distribution over $K$ experts – instead of 0: we minimize $\Omega^K_{\text{target}} = (\log(K) - \Omega_{\text{local}}(\mathbf{G}_m))^2$.

- *Merged entropy*: We sum the top $K$ and the bottom $N - K$ routing probabilities to give a binomial distribution, and optimize this to have entropy 0. This encourages the routing weight to all be in the top $K$ experts, but does not care exactly how it is distributed among them.

*BPR modifications* With these losses, the router uses $K > 1$ experts per token. However, BPR prioritises tokens according to their max routing probability, which decreases when probabilities are distributed over $K$ choices. The stabilisation effect BPR provides training is consequently lost. We alter it to prioritise tokens by the sum of top $K$ probabilities. In vision tasks, the two approaches perform identically [1], but here the latter stabilises training and unlocks $K > 1$.

Table 6 shows the final results. Without changes to the local entropy loss (BPR score = max, local entropy method = default), there are some improvements which stem from using $K > 1$ for image tokens - the local loss on text means it is effectively using $K = 1$ for text anyway. Without the modifications to the BPR score, the modifications to the local loss can result in fairly unstable models. Once the BPR score is modified, we see consistent improvements with increasing $K$, particularly with the 'Merged' variant.

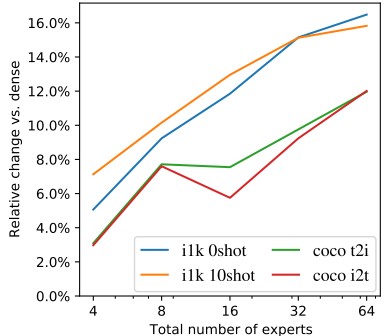

Figure 12: **Increasing the total number of experts** consistently improves model performance; all are better than the dense baseline.

### E.2 Increasing the total number of experts

There is thus far no consensus on the optimal number of experts in MoEs; early NLP research scaled to 1000s of experts [14, 8], before reducing to 32 or 64 [4], which is the standard setup for vision [1]. In Figure 12, we vary this for LIMoE, and show that larger expert pools yield consistent performance improvements.

Table 7: A simple routing setup without modality-specific adjustments is competitive with specialized approaches. 0shot and 10shot columns show accuracy (%), t2i and i2t show recall@1 (%).

| | routers | i1k 0shot | i1k 10shot | coco t2i | coco i2t |
|---|---|---|---|---|---|
| | joint | $56.9^{57.2}_{56.7}$ | $50.5^{52.0}_{49.0}$ | $25.5^{34.3}_{16.8}$ | $39.5^{51.8}_{27.1}$ |
| | per modality | 56.8 | 50.5 | 25.6 | 40.1 |
| disjoint experts (5 for text, 27 for image) | | 56.8 | 50.1 | 25.1 | 39.1 |

## E.3 Router design choice

Recall the router is simply a dense layer; by default we have a *joint* router for all tokens, independent of modality, with no constraints on gating. We consider two other options:

- *Per-modality router.* We consider modality-dependent routers which can leverage knowledge of token modality to improve performance (that is, one router for image tokens, and a different one for text tokens). They both output routing distributions over a shared pool of experts, similar to prior works have per-task routers for multitask learning [6].
- *Disjoint experts and routers.* We define separate pools of image and text experts. This way, image tokens can only go to a set of experts $\mathcal{E}_{\text{img}}$, and text tokens can only be assigned to another set of experts $\mathcal{E}_{\text{txt}}$. In principle, these sets may or may not intersect. In Table 7, we report results when the sets are indeed disjoint.

The results in Table 7 show the three approaches lead to comparable performance. In general, the disjoint setup was more stable, and did not need entropy regularisation as per-modality balance/independence is enforced by design. While convenient and well-behaved here, this approach may not be as general for the case with dozens of tasks and modalities.

## E.4 Pruning Multimodal Experts

During training we track what fraction of each modality's tokens went to each expert. It is therefore trivial to identify which experts are processing predominately text and which are processing predominately images. We show here that this information can be trivially used to prune experts for single-modality forward passes, demonstrating on two 32-expert LIMoE-S/16 models: one trained with global text entropy threshold $\tau_{\text{text}} = \log(4)$, and one with $\tau_{\text{text}} = \log(9)$.

**Choosing what to prune**. Note that we separately choose what experts to prune per-modality; we use text as an illustrative example. Pruning is simple: For each MoE layer, we rank experts according to the fraction of text tokens they processed during training (we average over the last 2500 steps with measurements sampled every 50 steps). We then start pruning according to the one that processed the least tokens, and so on. Figure 13 shows how the coverage of different modalities changes as experts are pruned. Following the relationship between the global text entropy threshold and the idea of the 'soft minimum', we see that around $e^{\tau_{\text{text}}}$ text experts are needed to process the majority of text tokens; e.g. with $\tau_{\text{text}} = 4$ for a single-modality forward pass, 28 experts could be comfortably pruned. Image experts are more distributed; almost all the experts are needed to process all image tokens, as expected.

**How to run LIMoE inference with fewer experts**. While some experts are pruned, the model is not further trained to adapt to this new situation. One must therefore think carefully on the best way to apply models with a subset of experts. The router predicts $p(\text{expert}|\mathbf{x})$. The top-$K$ experts are activated, and the output of the expert layer is the weighted average of the expert outputs. The weighting used for expert $i$ is the *unnormalized* $p(\text{expert}_i|\mathbf{x})$. This is important, as removing some of the experts and their logits modifies the concentration of $p(\text{expert}|\mathbf{x})$, and could result in expert weights higher than those used at training time.

When removing some of the experts, there are therefore two natural options:

1. router-drop: Completely remove the experts from the router. The softmax for $p(\text{expert}|\mathbf{x})$ will be computed over a subset of experts, thereby adjusting the weights as discussed above.

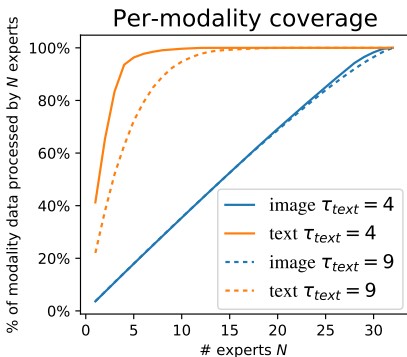

Figure 13: **Per-modality coverage**; for each modality, we progressively prune the least important experts. The coverage shows the percentage of router top-1 predictions which are still serviceable with the remaining experts. For text (orange), many experts can be pruned, but that is not the case for images. The global entropy threshold $\tau$ controls the prunability, as it encourages use of at least $\log(\tau)$ experts.

2. `router-pred`: The router still predicts probabilities for pruned experts. However, it is unable to actually use the pruned experts; the top-$K$ operation will ignore those that are unavailable. This preserves the original scaling the model was trained with.

The two approaches are naturally very similar if very few experts are removed. Illustrating with ImageNet-10shot (linear few-shot evaluation), Figure 14 compares the two options. When a large number of experts are pruned, the `router-pred` is significantly better, but if only a few experts are pruned, they both perform similarly.

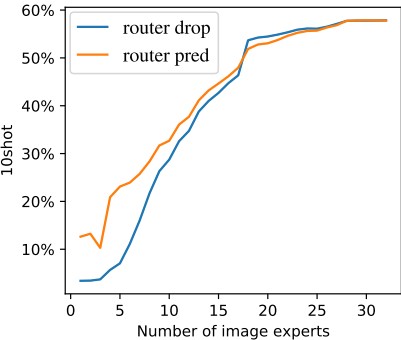

Figure 14: It is better to predict weights for pruned experts and mask them out *after* the softmax.

**The effect of pruning on performance** Figure 15 shows the impact of pruning image and text experts on zero-shot ImageNet accuracy. Recall that image and text inputs are processed independently for this evaluation, and so the experts used for each modality can be independently pruned.

As expected, we can prune down to only 4 experts during text evaluation without significantly harming performance. On the other hand, the less pruning of image experts, the better.

### E.5 Grouped routing

Splitting batches into groups before dispatching can reduce routing cost significantly, which depending on implementation can scale $\sim \mathcal{O}(\texttt{num tokens}^2)$. There are two sources of potential issues though: in our implementation, auxiliary losses are computed in each group then averaged. The necessary batch-wise statistics become less reliable with more numerous, smaller groups. Secondly, with smaller groups, it is more likely to get an almost homogenous batch, which makes distributing across

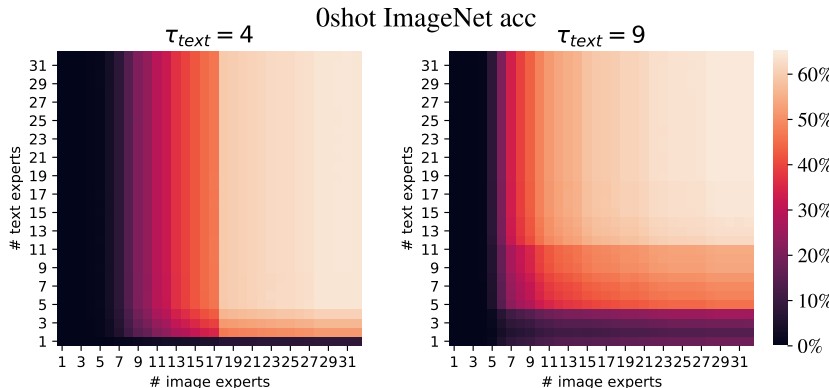

Figure 15: The impact of pruning on ImageNet zero-shot accuracy, comparing two LIMoE-S/16 models trained with different global text entropy thresholds.

Table 8: LIMoE performance on LAION400M, against a dense baseline, three trials.

|  | 10shot | 0shot | COCO t2i | COCO i2t |
|---|---|---|---|---|
| Dense | $47.7^{48.1}_{47.3}$ | $56.0^{56.3}_{55.6}$ | $27.8^{28.3}_{27.4}$ | $42.8^{43.4}_{42.1}$ |
| LIMoE | $54.3^{54.7}_{53.9}$ | $61.0^{61.4}_{60.7}$ | $28.9^{29.2}_{28.5}$ | $43.7^{44.6}_{42.9}$ |

experts harder. To study this, we sweep the group size in a parallel setup with 128 examples per device. Group size 1 means processing and dispatching $128 \times (196 + 16) = 27136$ tokens at once, whereas e.g. group size 8 involves splitting into 8 groups of 3392 tokens. Figure 16 shows the effect of this; up to 4 groups, performance is good, but any more than that and training becomes unstable, harming performance. This is more fragile than image-only routing, where group sizes as small as 400 are stable (equivalent to $\sim 68$ groups here). Nonetheless, with 4 groups, step time is reduced by 30%, capturing 75% of potential efficiency gains from grouped routing.

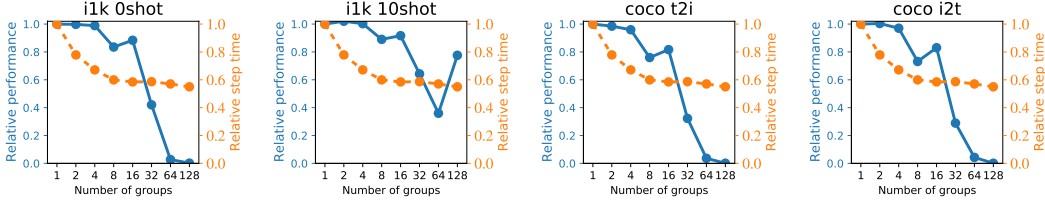

Figure 16: **Grouped routing can reduce step time (orange)**, but too much becomes unstable.

### E.6   Experiments on public data

In order to ascertain LIMoE's efficacy on public data, and reproducibility, we train B/16 models on LAION-400M [49]. We train for 5 epochs at batch size 16,384. Table 8 shows the outcome of three trials, compared against a dense baseline. Once again, we see significant improvements performance, especially in ImageNet zero-shot (+5.0% absolute, +8.9% relative) and 10-shot (+6.6% absolute, +13.8% relative) performance.

# F Model Analysis

## F.1 Routing Distributions

In this section, we explore how routing is distributed across different layers, experts, and modalities. In particular, we focus on which tokens are dropped. We analyze two models, B/32 and B/16, each with 8 experts. This way we can appreciate the impact of having a significantly different ratio of text:image tokens. Moreover, the global entropy targets $S = e^\tau$ for (text, image) tokens are (3, 25) and (6, 6) for the B/32 and B/16 models, respectively.

We first show the routing distributions under the training distribution in Figures 17 and 18. In both cases –as expected– routing works very well. Moreover, most experts handle both image and text tokens.

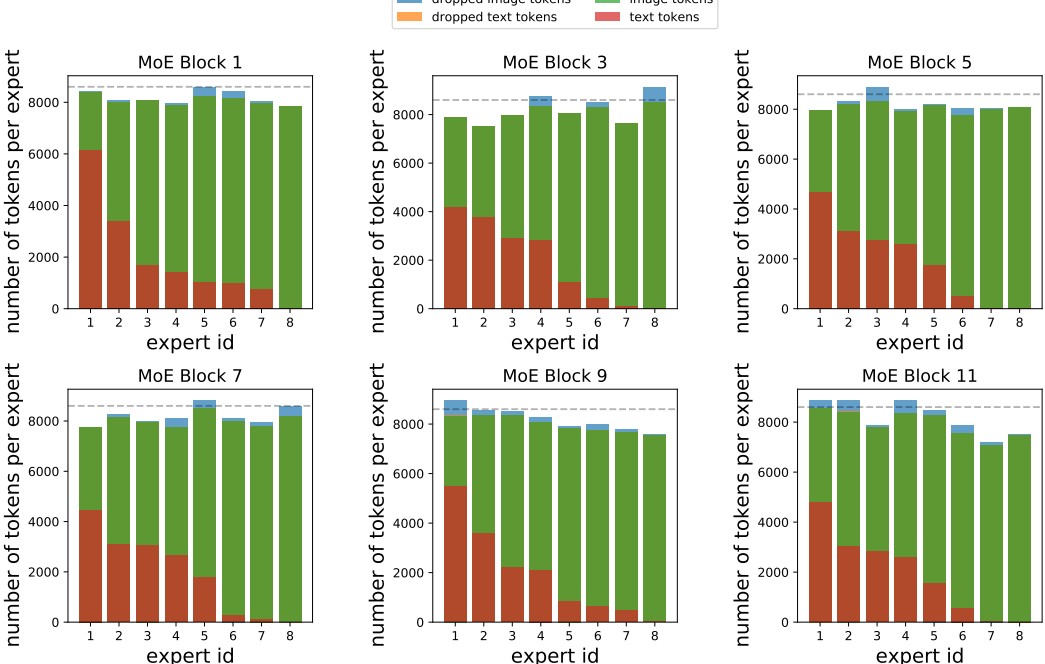

Figure 17: **Token Distribution for training data.** B/32 model with 8 experts. We display utilization and dropping for a forward pass with batch size 1024. The discontinuous line represents the maximum capacity per expert. Note that we enforce capacity locally per device, so some tokens may not be able to be dispatched even within global capacity constraints. We observe very little token dropping as this is the training data for which auxiliary losses lead to balance.

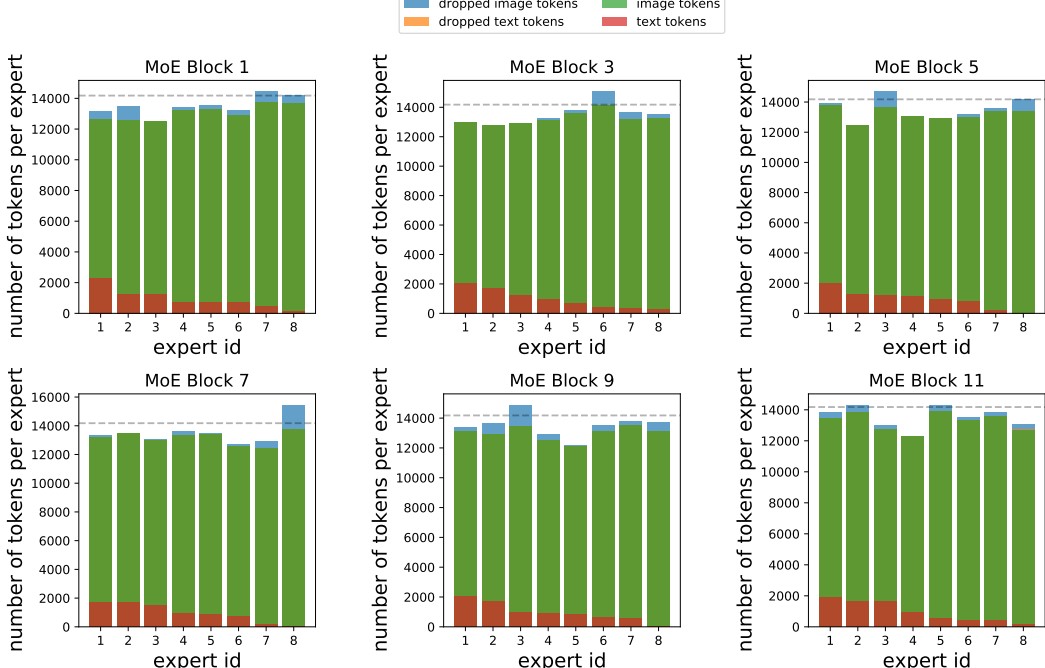

Figure 18: **Token Distribution for training data.** B/16 model with 8 experts. We display utilization and dropping for a forward pass with batch size 512. The discontinuous line represents the maximum capacity per expert. Note that we enforce capacity locally per device, so some tokens may not be able to be dispatched even within global capacity constraints. We observe very little token dropping as this is the training data for which auxiliary losses lead to balance. Compared to Figure 17, we can see how text tokens generally represent a quite small fraction of the in-flow for every expert.

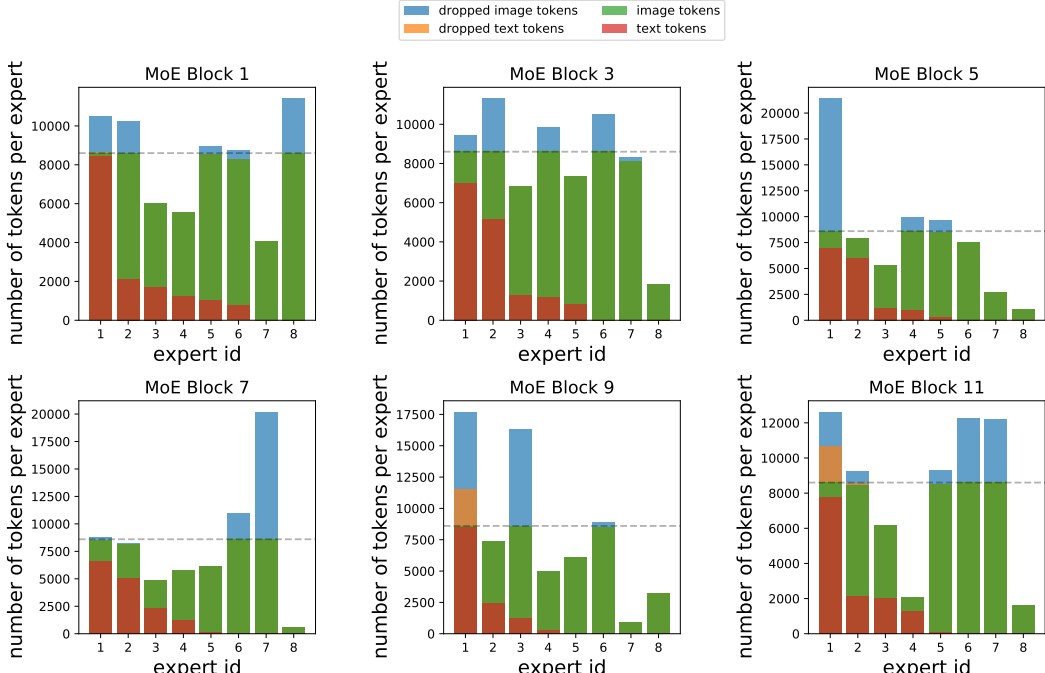

Figure 19: **Token Distribution for COCO data.** B/32 model with 8 experts. We display utilization and dropping for a forward pass with batch size 1024. The discontinuous line represents the maximum capacity per expert. Note that we enforce capacity locally per device, so some tokens may not be able to be dispatched even within global capacity constraints. Compared to Figure 17, in this case, as there is a distribution shift –while no further training or finetuning–, we see distributions of tokens per expert becoming fairly unbalanced. Moreover, a non-trivial amount of tokens are dropped (above discontinuous horizontal line). Even text tokens are dropped sometimes, and some experts –like Expert 1 in the MoE Block 1 or 9– end up only processing text tokens.

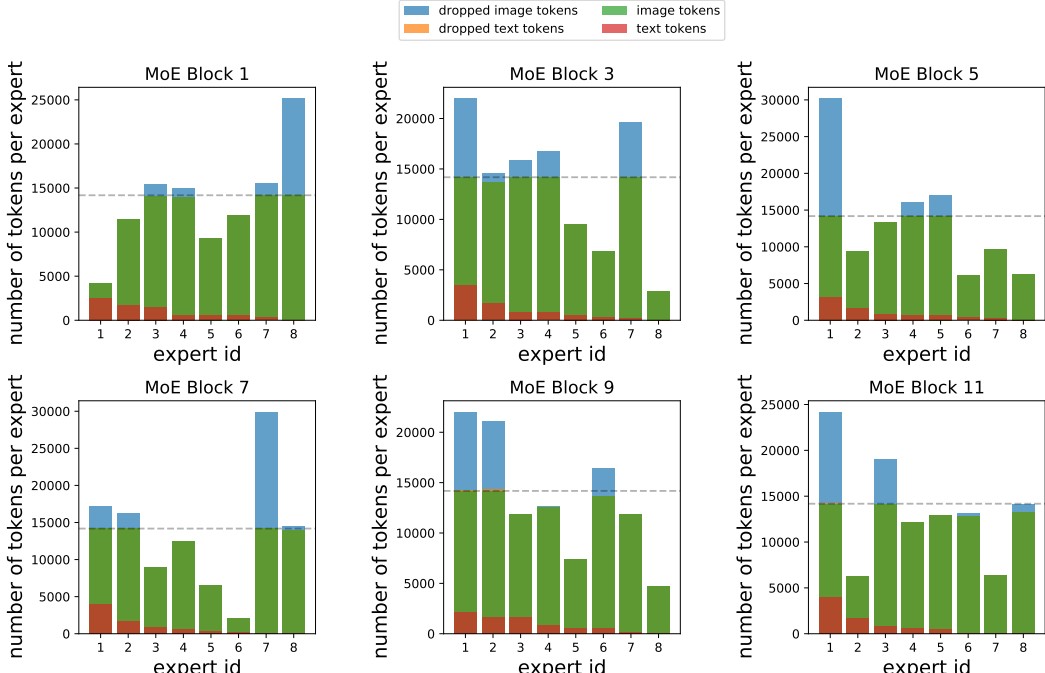

Figure 20: **Token Distribution for COCO data.** B/16 model with 8 experts. We display utilization and dropping for a forward pass with batch size 512. The discontinuous line represents the maximum capacity per expert. Note that we enforce capacity locally per device, so some tokens may not be able to be dispatched even within global capacity constraints. Compared to Figure 18, in this case, as there is a distribution shift –while no further training or finetuning–, we see distributions of tokens per expert becoming fairly unbalanced. Moreover, a non-trivial amount of tokens are dropped (above discontinuous horizontal line). Text tokens are still mostly processed as BPR shields them via their high priorities and they still represent a small percentage of the tokens.

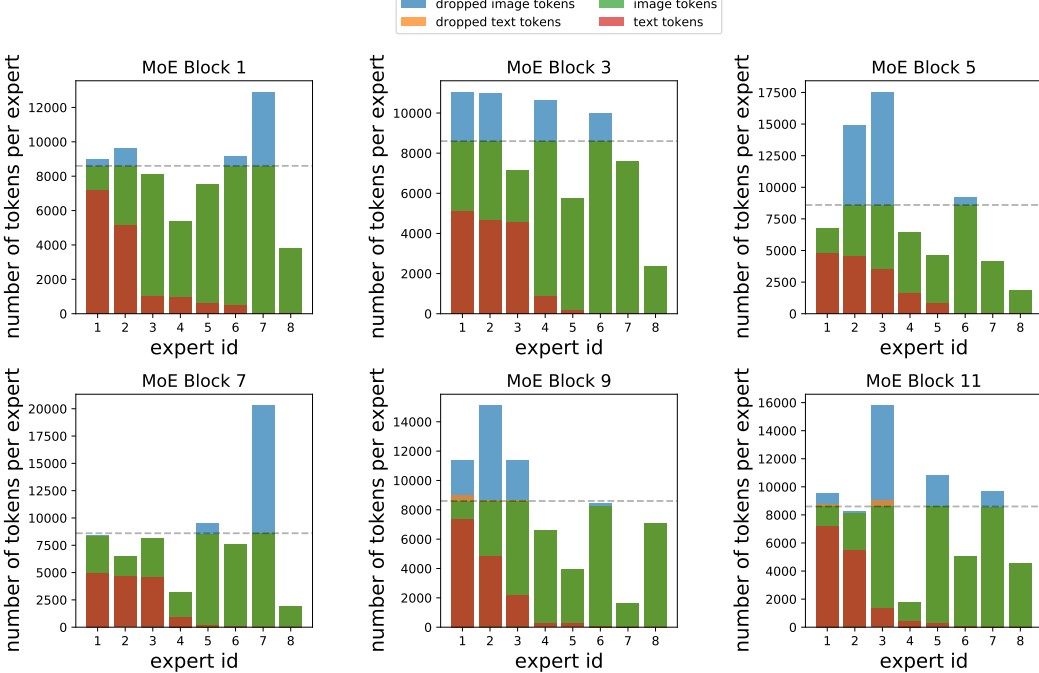

Figure 21: **Token Distribution for ImageNet data.** B/32 model with 8 experts. We display utilization and dropping for a forward pass with batch size 1024. The discontinuous line represents the maximum capacity per expert. Note that we enforce capacity locally per device, so some tokens may not be able to be dispatched even within global capacity constraints. Compared to Figure 17, in this case, as there is a distribution shift –while no further training or finetuning–, we see distributions of tokens per expert becoming fairly unbalanced. Moreover, a non-trivial amount of tokens are dropped (above discontinuous horizontal line). Very few text tokens are dropped (there is a significant amount of padding, and prompt tokens that are probably processed with very high confidence scores by BPR).

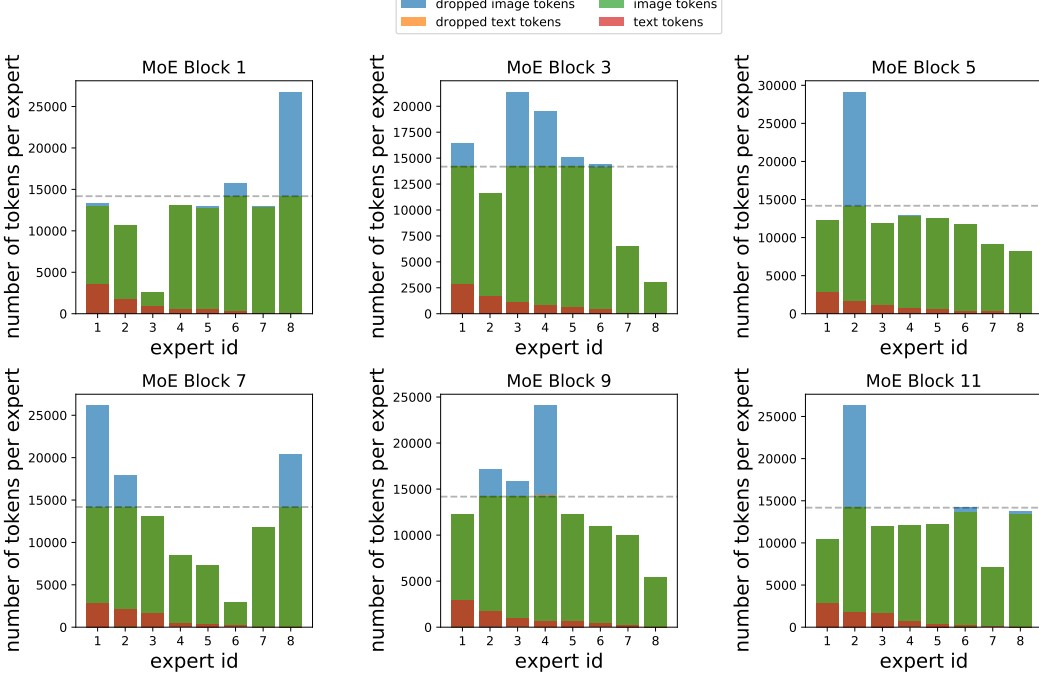

Figure 22: **Token Distribution for ImageNet data.** B/16 model with 8 experts. We display utilization and dropping for a forward pass with batch size 512. The discontinuous line represents the maximum capacity per expert. Note that we enforce capacity locally per device, so some tokens may not be able to be dispatched even within global capacity constraints. Compared to Figure 18, in this case, as there is a distribution shift –while no further training or finetuning–, we see distributions of tokens per expert becoming fairly unbalanced. Moreover, a non-trivial amount of tokens are dropped (above discontinuous horizontal line). Almost no text tokens are dropped (there is a significant amount of padding, and prompt tokens that are probably processed with very high confidence scores by BPR).

## F.2 Routing Examples

In this section, we share practical examples of image and text token routing on the B/32 and B/16 models introduced at the beginning of the section. All evaluations are on ImageNet (that is, not on the training data). While the number of experts is clearly smaller than the number of different semantic concepts in images and text, we still highlight some cool patterns in most experts – especially in the context of images, as text tokens tend to use a reduced number of experts. We show some of the patches with the highest routing confidence, as analyzing all the thousands of patches that are assigned to each expert is difficult. However, we expect many other semantic concepts present in the training data to be almost exclusively served by individual experts.

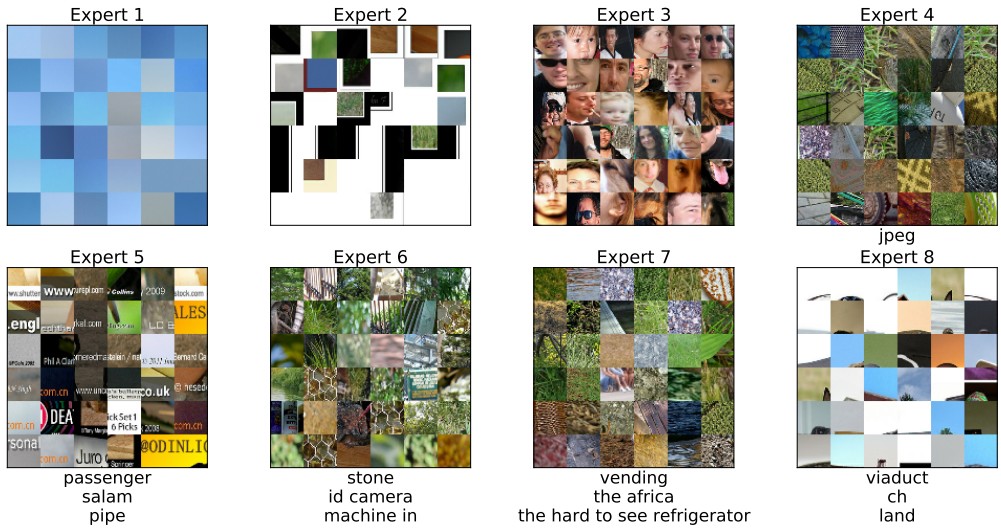

Figure 23: **Token routing for Imagenet.** B/32 model with 8 experts, we show some of the original tokens (both image and text) as routed at the second MoE layer (corresponds to the fourth encoder).

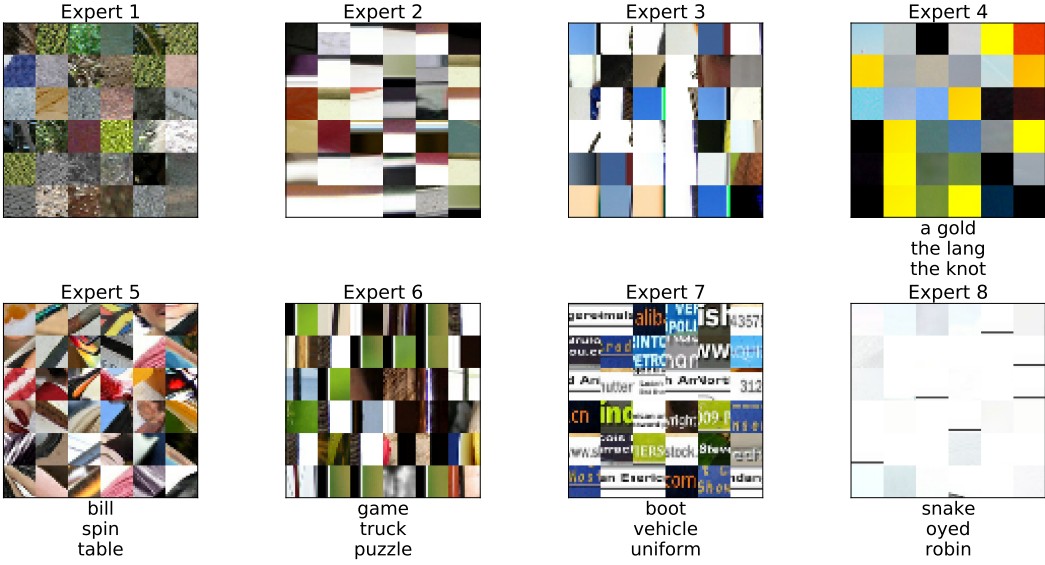

Figure 24: **Token routing for Imagenet.** B/16 model with 8 experts, we show original tokens (both image and text) as routed at the first MoE layer (corresponds to the second encoder block).

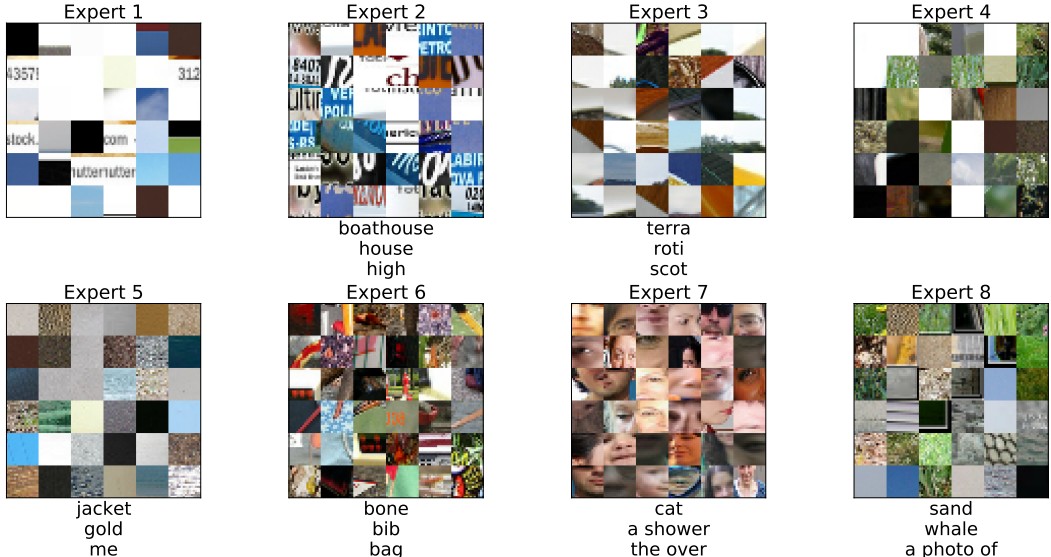

Figure 25: **Token routing for Imagenet.** B/16 model with 8 experts, we show original tokens (both image and text) as routed at the second MoE layer (corresponding to the fourth encoder block).

## F.3 Routing for Individual Inputs

In this subsection, we show the expert split for a specific given input – image and text. Recall tokens from different modalities do not interact in the forward pass (other than via sharing expert capacity).

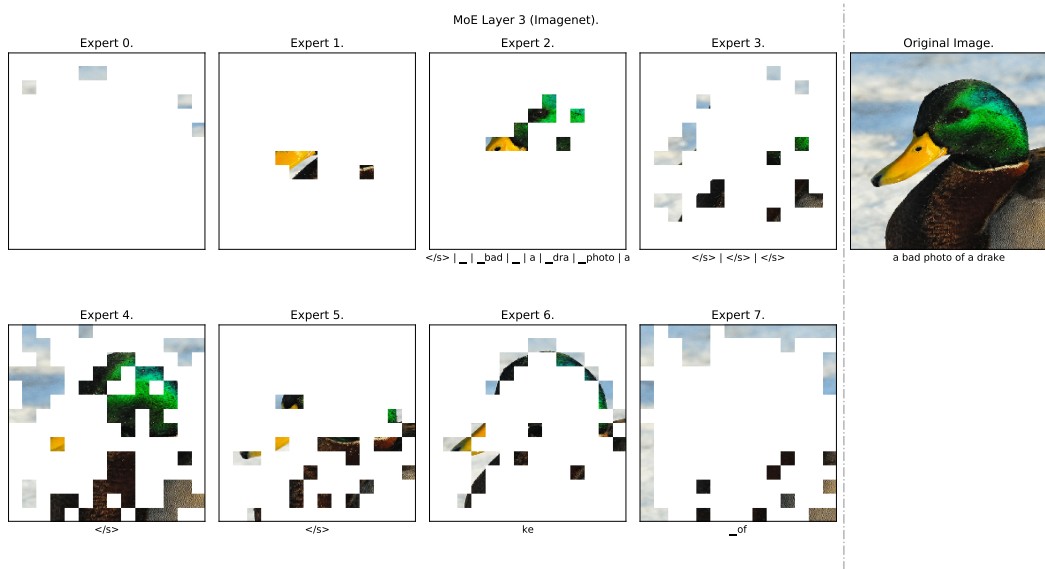

Figure 26: **Token routing for an Imagenet input.** B/16 model with 8 experts, we show original tokens (both image and text) as routed at the second MoE layer (corresponding to the fourth encoder block, while we use zero-indexing). The original image and text are displayed on the right-hand side.

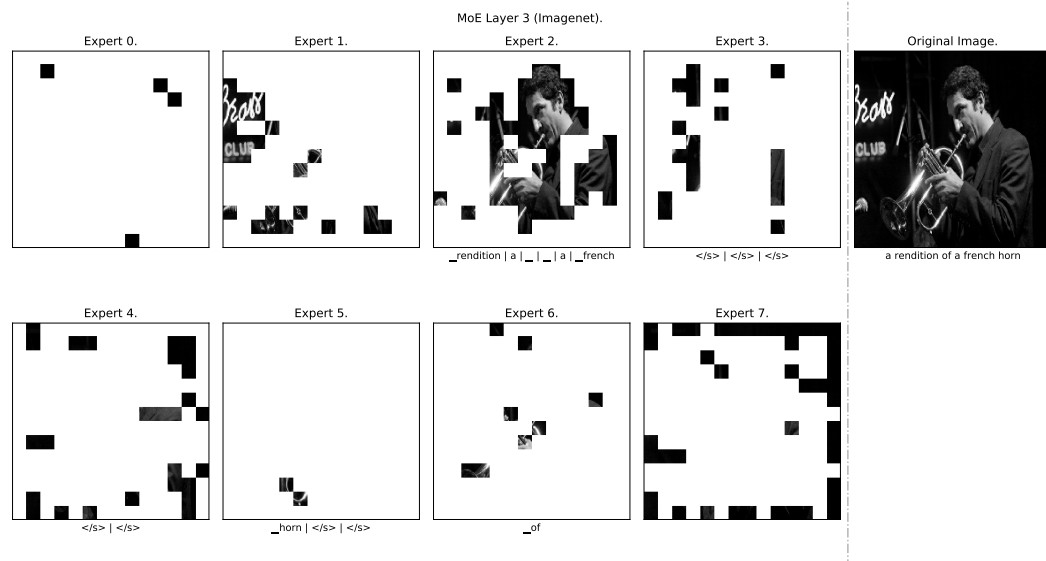

Figure 27: **Token routing for an Imagenet input.** B/16 model with 8 experts, we show original tokens (both image and text) as routed at the second MoE layer (corresponding to the fourth encoder block, while we use zero-indexing). The original image and text are displayed on the right-hand side.

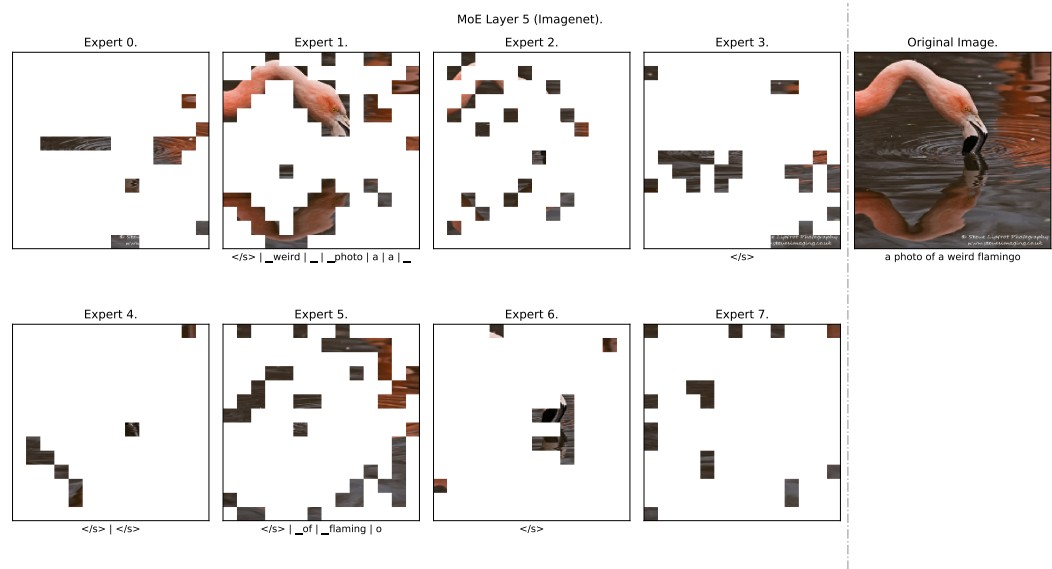

Figure 28: **Token routing for an Imagenet input.** B/16 model with 8 experts, we show original tokens (both image and text) as routed at the third MoE layer (corresponding to the sixth encoder block). The original image and text are displayed on the right-hand side.

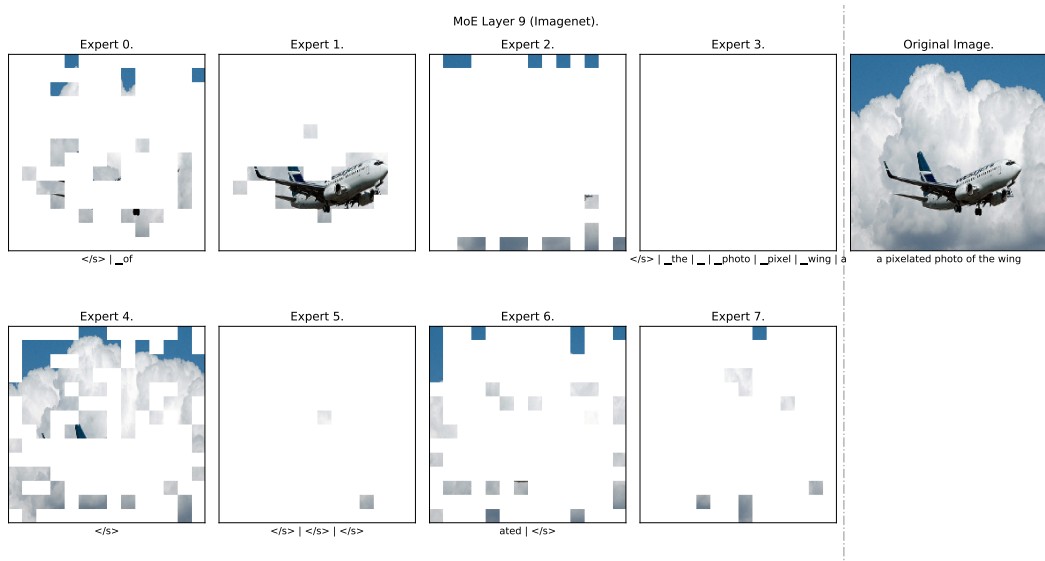

Figure 29: **Token routing for an Imagenet input.** B/16 model with 8 experts, we show original tokens (both image and text) as routed at the previous-to-last MoE layer (corresponding to the tenth encoder block, while we use zero-indexing). The original image and text are displayed on the right-hand side.

## F.4 Routing Trajectories

In this section, we try to have a look at the overall trajectories followed by both image and text tokens across the network. While definitely a complex endeavor, we show in Figure 30 for B/32 and Figure 31 for B/16 the main trajectories followed by such tokens. Interestingly enough, it seems that for both models and image tokens, the first two/three MoE layers are fairly interconnected – in other words, given the expert selected for some token in one layer, it may be hard to predict the next steps. Text tokens (probably given that very few experts are indeed often used for text) have more predictable trajectories.

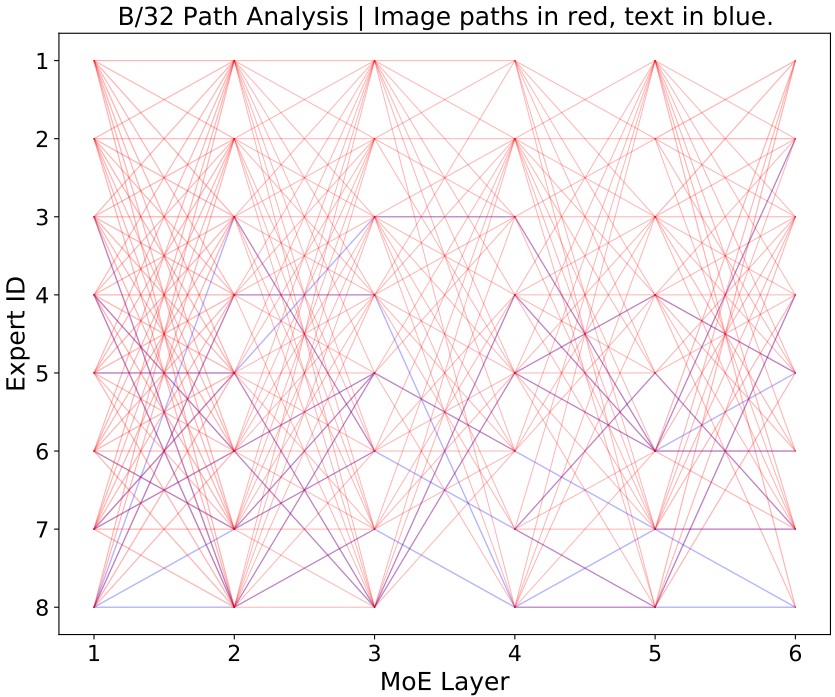

Figure 30: **Token trajectories.** B/32 model with 8 experts, we show the main expert-routes followed by text tokens (in blue) and image tokens (in red).

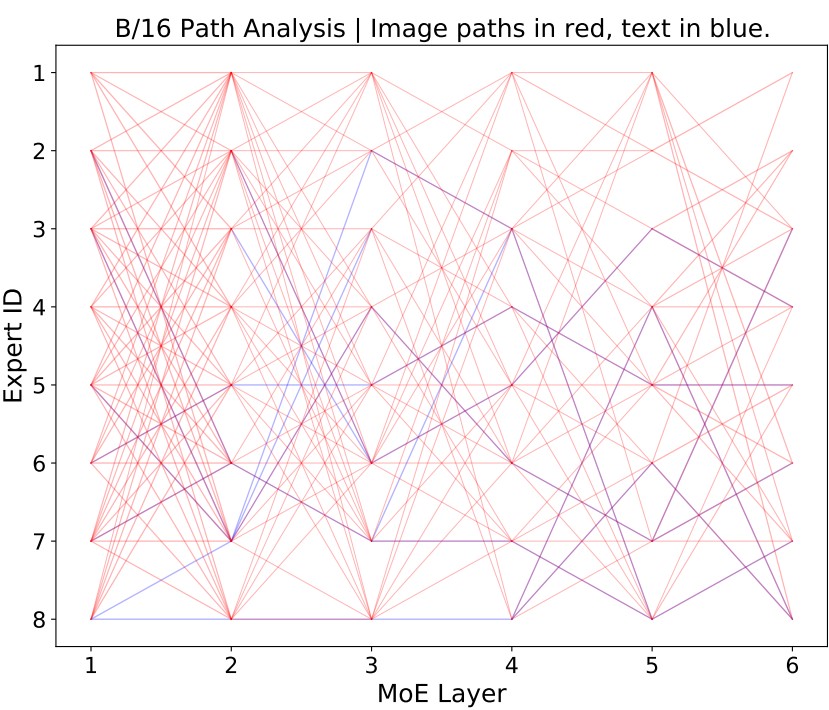

Figure 31: **Token trajectories.** B/16 model with 8 experts, we show the main expert-routes followed by text tokens (in blue) and image tokens (in red).

### F.5 BPR rankings

The local entropy loss encourages concentrated routing predictions with high $p_{\max}$ for text. At the same time, BPR prioritises tokens with high $p_{\max}$. One might assume that this combination is effectively just ranking all text tokens first. The following plots give us some insight into how the buffers end up sorting tokens from both modalities. Figures 32 and 33 show the priority distribution on the training data for the B/32 and B/16 models, respectively. Under a data shift, Figures 34 and 35 show the same statistics for COCO data, and Figures 36 and 37 for ImageNet. In these cases, no extra training was performed (i.e., it is zero-shot). Overall, we see that while text tokens enjoy by default a much higher priority, this is not always the case, and some (important?) image patches are sometimes processed before other text tokens.

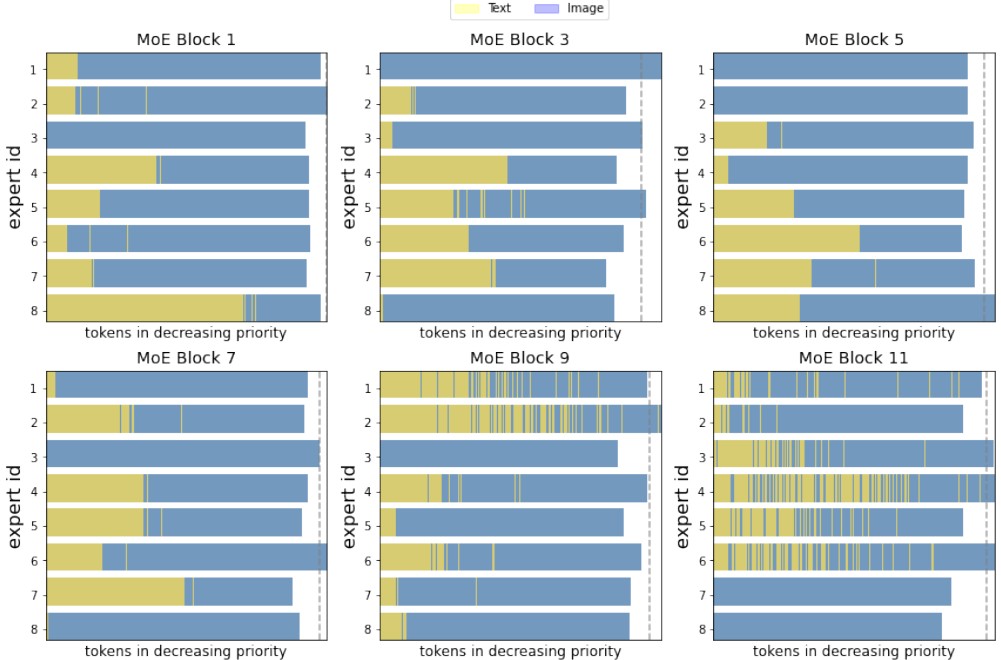

Figure 32: **Token priorities for training data.** B/32 model with 8 experts. We see that –especially in later layers– token priorities are mingled across modalities, whereas text tokens tend to have higher scores (and, thus, BPR priorities). Tokens to the left of the $x$-axis are given more priority. The vertical discontinuous line corresponds to the per-expert global capacity limit. Tokens beyond that point are not processed by the expert.

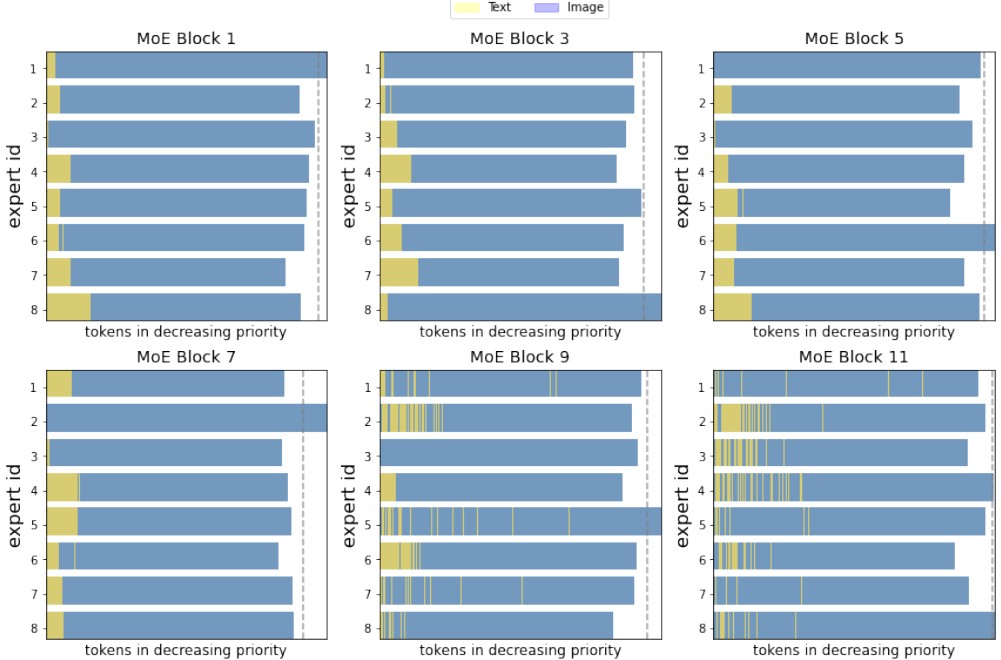

Figure 33: **Token priorities for training data.** B/16 model with 8 experts. We see that –especially in later layers– token priorities are mingled across modalities, whereas text tokens tend to have higher scores (and, thus, BPR priorities). Compared to the B/32 model, here we see a longer tail of low-priority image tokens. Tokens to the left of the $x$-axis are given more priority. The vertical discontinuous line corresponds to the per-expert global capacity limit. Tokens beyond that point are not processed by the expert.

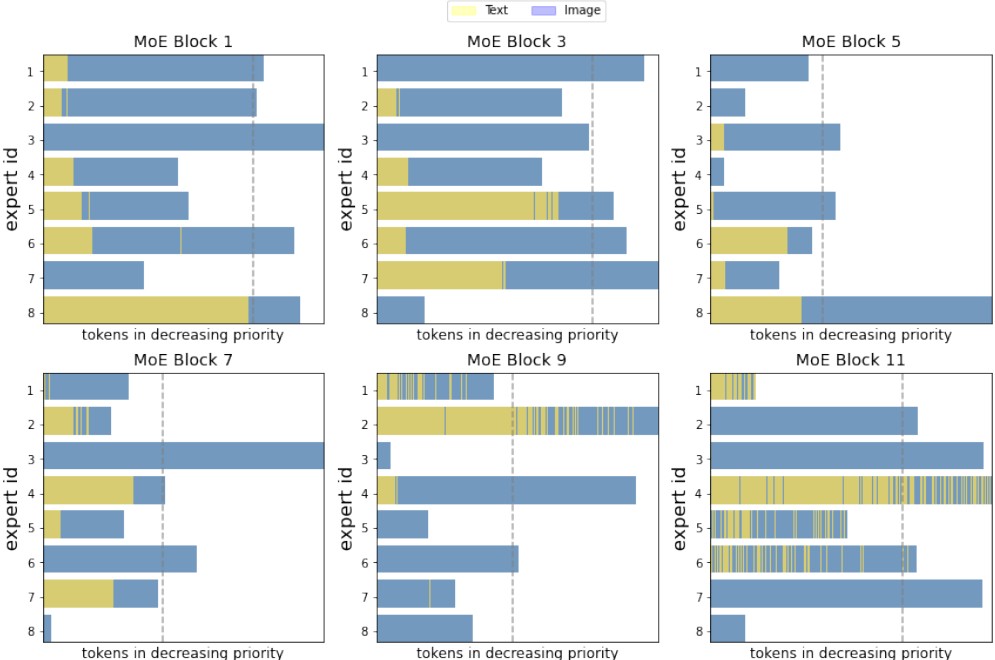

Figure 34: **Token priorities for COCO data.** B/32 model with 8 experts. We see that –especially in later layers– token priorities are mingled across modalities, whereas text tokens tend to have higher scores (and, thus, BPR priorities). Tokens to the left of the $x$-axis are given more priority. The vertical discontinuous line corresponds to the per-expert global capacity limit. Tokens beyond that point are not processed by the expert. Due to the distribution shift (this is evaluated on COCO, which was not the training data), we see lots of dropping is actually happening (mostly images, but also some text tokens).

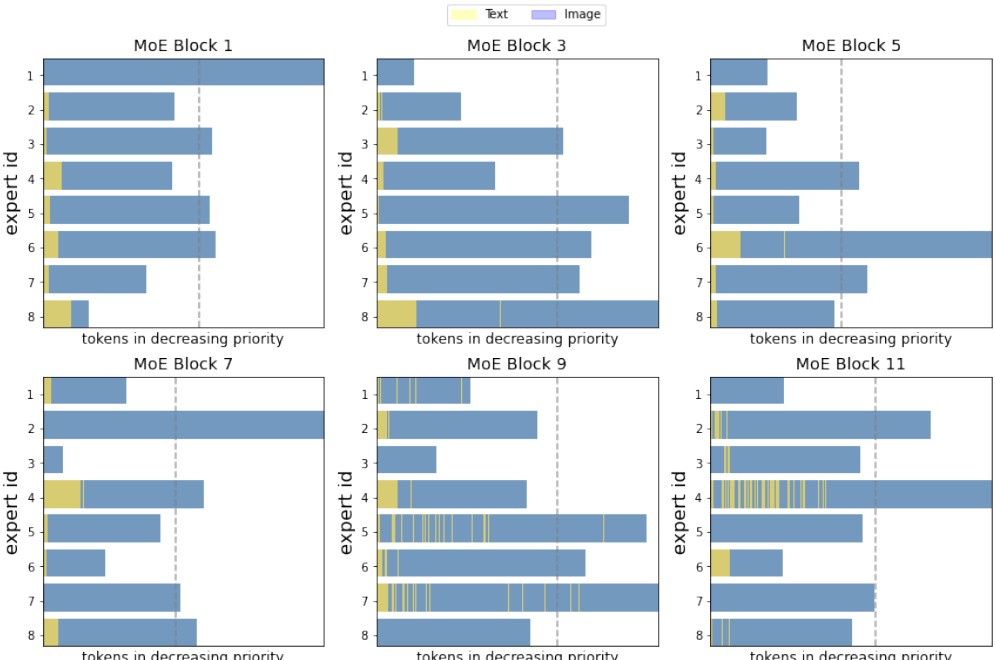

Figure 35: **Token priorities for COCO data.** B/16 model with 8 experts. We see that –especially in later layers– token priorities are mingled across modalities, whereas text tokens tend to have higher scores (and, thus, BPR priorities). Tokens to the left of the $x$-axis are given more priority. The vertical discontinuous line corresponds to the per-expert global capacity limit. Tokens beyond that point are not processed by the expert. Due to the distribution shift (this is evaluated on COCO, which was not the training data), we see lots of dropping is actually happening (while pretty much only image tokens).

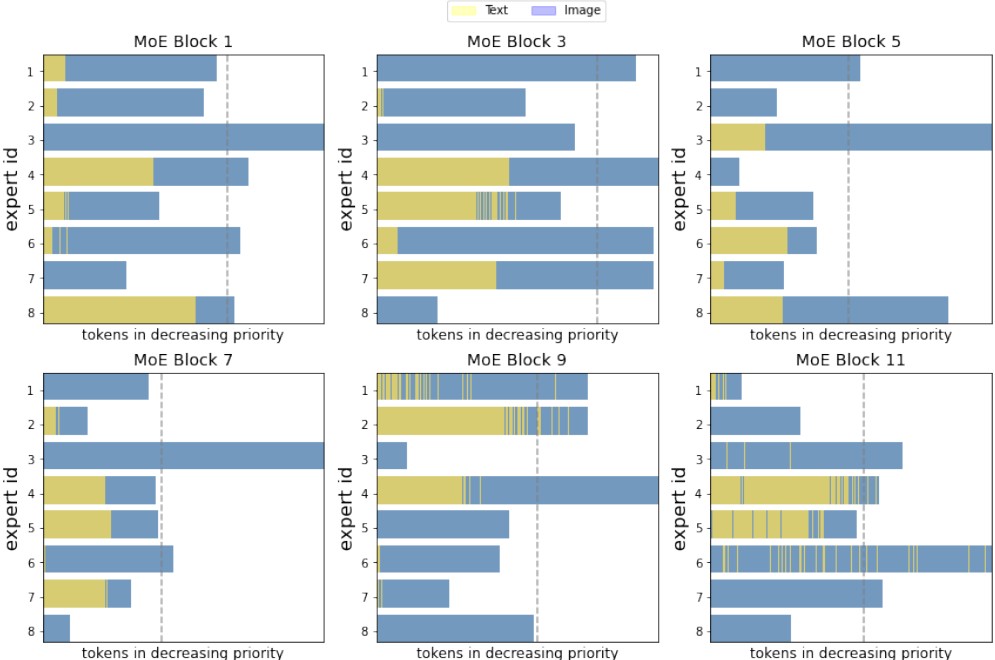

Figure 36: **Token priorities for ImageNet data.** B/32 model with 8 experts. We see that –especially in later layers– token priorities are mingled across modalities, whereas text tokens tend to have higher scores (and, thus, BPR priorities). Tokens to the left of the $x$-axis are given more priority. The vertical discontinuous line corresponds to the per-expert global capacity limit. Tokens beyond that point are not processed by the expert. Due to the distribution shift (this is evaluated on ImageNet, which was not the training data), we see lots of dropping is actually happening (mostly images, but also some text tokens).

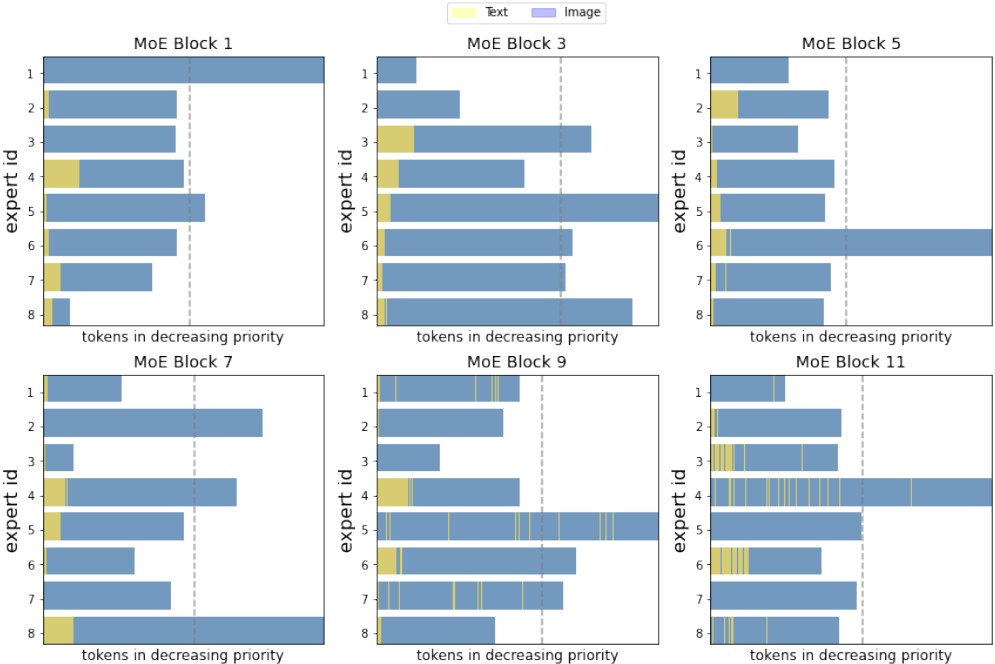

Figure 37: **Token priorities for ImageNet data.** B/16 model with 8 experts. We see that –especially in later layers– token priorities are mingled across modalities, whereas text tokens tend to have higher scores (and, thus, BPR priorities). Tokens to the left of the $x$-axis are given more priority. The vertical discontinuous line corresponds to the per-expert global capacity limit. Tokens beyond that point are not processed by the expert. Due to the distribution shift (this is evaluated on ImageNet, which was not the training data), we see lots of dropping is actually happening (mostly images, but also some text tokens).

# G LIMoE-H/14 Analysis

In this section, we share some details and analysis regarding our largest model, the LIMoE-H/14. Figure 38 shows the development of the max routing probability across different MoE layers. Figure 2 shows qualitatively the specialization of image experts. Experts naturally specializing on semantic concepts such as body parts (hands, eyes), textures, fauna, food and doors. In Figure 39, we show the distribution of tokens per type and expert for every layer. Note that we set the entropy loss to approximately require at least 4 text experts, something that seems to agree well with the plot (in this case the ratio text:image tokens was close to 1:27).

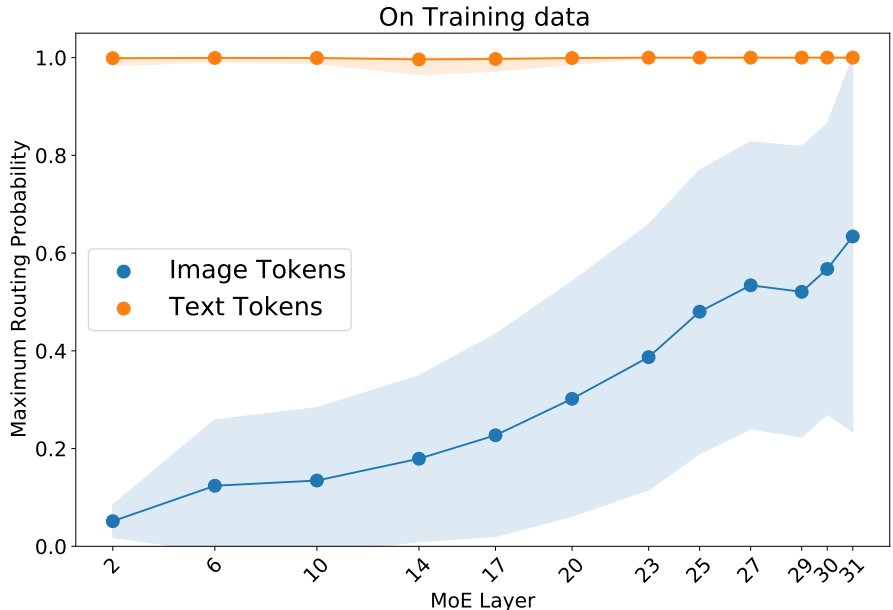

Figure 38: **Per-token $p_{max}$ distribution for training data.** For LIMoE-H/14 model, we show the average and one standard deviation of the per-token maximum routing probability (corresponding to the selected expert). We see that for image tokens the model is increasingly confident, whereas for text tokens –given the local entropy loss– most of the predictions are close to one-hot.

## G.1 Preliminary analysis of text routings

We analyse the routing distributions of text tokens for LIMoE-H/14, using NLTK [50] to distinguish between verbs, nouns, adjectives, prepositions and determiners. Note that the SentencePiece tokenizer breaks words into smaller units, which are not necessarily always handled by the same expert, so it is not possible to perfectly parse every token processed by every expert.

The majority of tokens are from images, so only 3-4 experts handle text in this scenario. Figure 40 contains preliminary analysis, showing for each expert the breakdown of tokens it handles. Though some experts process a bit of everything (e.g. experts 0 and 1 in layer 6 and 31), there are signs of some semantic specialization. There are often experts which process mostly padding tokens. In Layer 14, expert 1 processes no prepositions, determiners or verbs, focussing on nouns and adjectives (and some padding); similarly expert 1 processes very few nouns or adjectives, instead handling padding tokens.

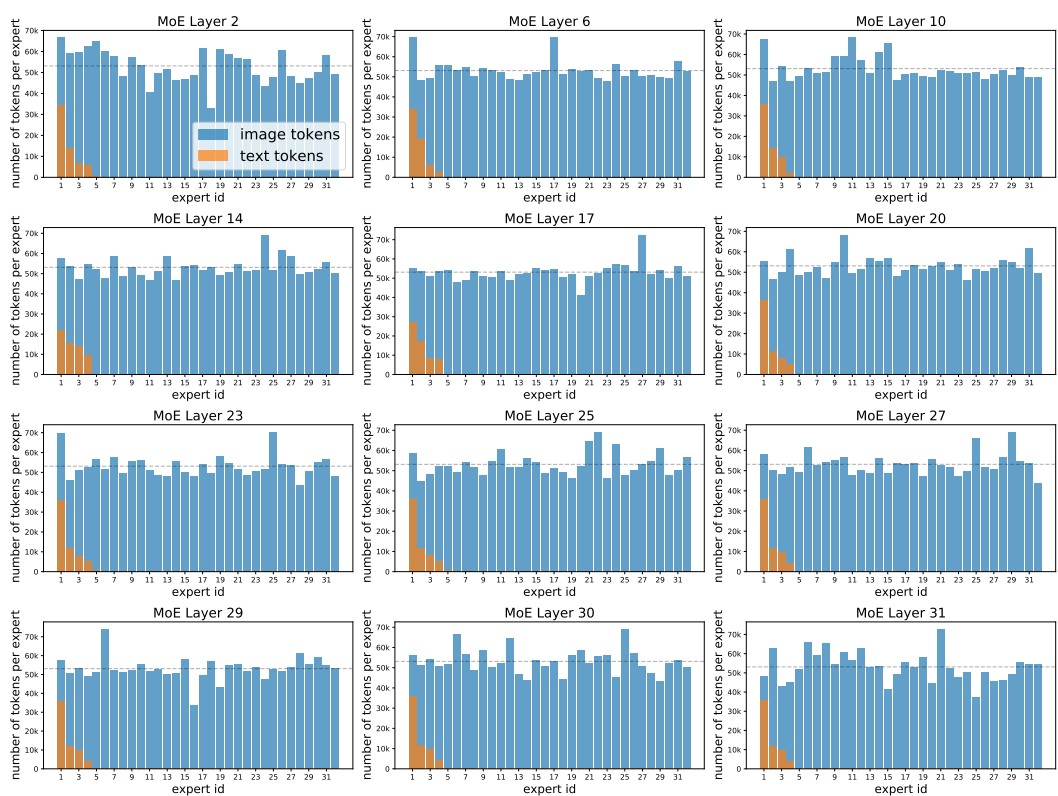

Figure 39: **Token routing per expert for** `LIMoE`**-H/14.** We show for each MoE layer and expert, the number of tokens per modality that were routed in a number of forward passes from the training data. When above the expert capacity (discontinuous horizontal line), some tokens were dropped – but not necessarily the image ones; for simplicity, we always show image tokens on top of text ones. In this setup, the ratio text:image tokens was close to 1:27.

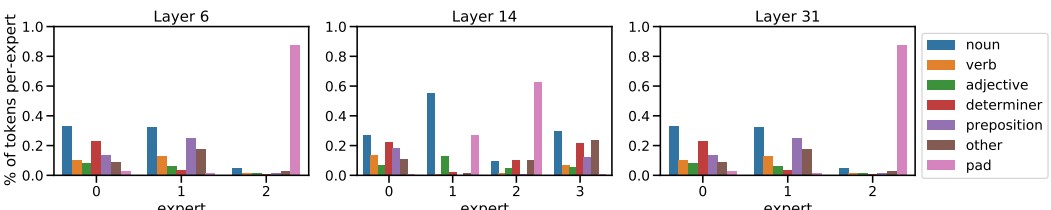

Figure 40: **Analysis of text routing for** `LIMoE`**-H/14**