# OpenReview forum: "Multimodal Contrastive Learning with LIMoE: the Language-Image Mixture of Experts"
_NeurIPS.cc/2022/Conference — NeurIPS 2022 Accept_

### Official Review · Reviewer_Lcgn · 2022-07-11

**Rating:** 7
**Confidence:** 5
**Soundness:** 4 excellent
**Presentation:** 4 excellent
**Contribution:** 3 good

**Summary:**

This paper proposes a new multimodal contrastive learning framework named as LIMoE. This is the first large-scale mixture-of-experts model for multiple modalities. Specifically, this paper designs a modality agnostic model which is not explicitly conditioned on modality. To address the training stability issue and to balance expert utilization, two entropy-based regularization losses are introduced. The effectiveness of this proposed model is quantitatively and qualitatively evaluated across multiple scales and network architectures.

**Questions:**

1. Table 2: the caption contains two “for” which should be a typo
2. It is suggested to use “zero-shot” rather than “0shot”


**Strengths And Weaknesses:**

---

Originality: the design of the multimodal mixture-of-experts (MoE) model is novel. To the best of my knowledge, previous MoE models are designed for single modality. The two regularization losses are widely used in previous studies, but the combination with MoE is valuable.

---

Quality:

Strengths: this work is technically sound. The claims are well supported by comprehensive empirical studies. The proposed regularization losses are reasonable and are expected to stabilize the training, and in turn bring performance improvement.

Weakness: The two proposed losses (as shown in Equation 2), which are also known as information maximization loss, are widely used in previous studies. It is quite straightforward to use this loss to mitigate the collapse issue in expert utilization. Therefore, this work is more like an experimental study.

---

Clarity: this paper is well written and well organized. It is easy to follow.

Significance: this work is a good effort in introducing MoE into multimodal contrastive learning. Although the proposed solution is trivial, it brings some insights to the community.

---

---

> ### Author Response · Authors · 2022-08-02
> **Initial author response**
>
> Many thanks for the time spent reviewing and your feedback, and for noting those typos/wording suggestions - we will amend the text accordingly.
>
> Regarding the losses, there are some prior works (e.g.[1]) that use entropy based approaches to regularise expert routing distributions. We don’t know of any prior works that specifically use mutual information, or use entropy to regularise multimodal MoE models.
>
> We discussed the connection between the global/local loss combination, and the mutual information (end of Section 2.2.2). Note that due to the threshold used on the global loss, which enables some modality-wise specialization of experts, we are not actually using the mutual information loss. In section 4.1 we present results which do use the mutual information loss (i.e. local + global entropy without threshold) and found the performance was worse at B/16 scale. We also found it did not scale stably to L/16.
>
> Regarding whether this solution is trivial, we first note that these auxiliary losses were not the only key of the solution, with other aspects such as Batch Priority Routing being key (see Figure 5). Given the modification to mutual information, the lack of proceeding works using it as an auxiliary loss (and indeed any works which study multimodal per-token routing at all), and the other aspects of our proposed solution, we believe our approach is indeed novel and non-trivial.
>
> As a final note, if you know of any prior works which use mutual information or entropy to regularise per-example conditional computation models, we would love to read them & cite them accordingly!
>
> [1]Multi-Source Domain Adaptation with Mixture of Experts, J. Guo, D. J. Shah, R. Barzilay, [arxiv:1809.02256](https://arxiv.org/abs/1809.02256)

---

### Official Review · Reviewer_g9Jj · 2022-07-11

**Rating:** 6
**Confidence:** 3
**Soundness:** 3 good
**Presentation:** 3 good
**Contribution:** 3 good

**Summary:**

The paper addresses an important problem, in which multimodal data is required for a model, by designing a unified architecture that accepts images and text concurrently with the use of the contrastive learning method for multimodal representation, trained on a paired Image-Text dataset. In addition, the paper addresses load balancing, which is one of the most important problems in the MoE setting, with an entropy-based regularization technique.

**Questions:**

I am not sure if I miss/misunderstand something including the Strengths/Weaknesses above. Still, it is not clear to me how the routing for text and images is done.  Using image-text pairs, it is said that it’s possible that all text can go to a single expert while image tokens are distributed almost equally. So they are routed not in pairs but individually (which is somewhat not similar to the illustration in Figure 1). Please explain why and why not are you separate the 2 modalities at routing?


Another somewhat related question is that for contrastive training, how do you identify the negative examples?


**Ethics Review Area:**

["I don’t know"]

**Limitations:**


Maybe not very relevant since the paper addresses the system-related level and thus is hard to judge those impacts.


**Strengths And Weaknesses:**

Strength:
1. The paper is well written.
2. The paper offers many studies that are useful for understanding many aspects it mentions.

Weakness:
1. The paper claims to be the first multimodal sparse MoE model, which is probably incorrect/too strong (line 299), e.g. [0] was first submitted in Nov 2021.
2. The fact that this paper centers around image-text data leads to the need of showing results of image-text benchmarks e.g. in [0], while the paper focuses on zero-shot. Even so why the main results in Table 1 doesn’t have other results for other scales of the model (likewise, if you have an even larger model having comparable parameters, would it beat the performances of those baselines?). For tasks other than Imagenet classification, it would be more convincing if the paper presents the comparison with some current baselines, in addition to the ablations.
3. Some acronyms might need further explanation such as the ones in Figure 3 for different models, e.g. B.16, S.16, …
4. No code is provided.


[0] VLMO: Unified Vision-Language Pre-Training with Mixture-of-Modality-Experts (2111.02358.pdf (arxiv.org)

---

> ### Author Response · Authors · 2022-08-02
> **Initial author response**
>
> Many thanks for the time spent reviewing the paper. We would first like to clarify the nature of LIMoE - our paper specifically studies learned conditional computation. This relates to your question `how the routing for text and images is done`: For LIMoE, and preceding works in NLP and Vision,  examples are embedded to a sequence of tokens, as per typical transformer-based approaches. In the FFN layer, instead of a single FFN, we have $N$ FFNs - each an “expert”. Learned routers decide which experts should process which tokens. The expert is a simple dense layer which, combined with a softmax, predicts from each token’s representation a routing distribution over $N$ experts.
>
> For LIMoE, the tokens can be either image or text examples. You are precisely correct that we do not separate the modalities for routing, and do not route in pairs*. The router just sees a big bag of tokens which need to go to experts - it doesn’t know which example it comes from or what modality that example is. We were motivated to study this as we believe this modality-agnostic approach will scale better as one increasingly adds modalities. Table 7 in the appendix shows an ablation with two alternative variants: where either the router knows what kind of token it is handling (text or image) or we directly have two independent routers, one per modality. None of the above clearly outperformed our simpler and more general approach.
>
> **You mentioned that Figure 1 does not show clearly that routing is done independently; if you have suggestions on how it can be clarified and improved, we would appreciate it!*
>
> We will now address some of the weaknesses/questions:
>
>
> 1. `The paper claims to be the first multimodal sparse MoE model, which is probably incorrect/too strong (line 299), e.g. [0] was first submitted in Nov 2021.`
> * Thanks for pointing us towards this paper! We will definitely add it as a reference, especially for one-tower models for contrastive learning for which there is very little literature.
> * As noted, we specifically study learned conditional computation, similar to e.g. the Switch Transformer or V-MoE. Though the referenced paper has per-modality FFNs, the structure is otherwise fundamentally different to the class of models we develop here: the FFN is decided in advance for each modality, and all tokens from a given example will always go to the same FFN based on its modality. It is also not sparse; there is no way to scale up the number of experts while keeping computational cost constant. In contrast, LIMoE’s routers learn to assign experts to each token.
> * Overall, VLMO does not have experts, sparsity or routing of the type studied here or developed by relevant prior works. This style of conditional computation is often (somewhat ambiguously) called “sparse MoEs”, hence why the claim may seem too strong; we will update wording in a camera ready version to make this distinction clearer.
>
> 2. `The fact that this paper centers around image-text data leads to the need of showing results of image-text benchmarks e.g. in [0], while the paper focuses on zero-shot`
> * We first note that zero-shot classification as studied in this work and prior work does necessitate good image *and* language understanding. Furthermore, we also present results on COCO image-text retrieval, a classic image-text task, similar to the referenced paper and prior works. Though we do not finetune on downstream tasks, zero-shot evaluation is sufficient to evaluate the models’ ability to learn multimodal representations.
>
> * `Even so why the main results in Table 1 doesn’t have other results for other scales of the model`  We study scale quite extensively in Figure 3 (same results also provided  in Table 4 of the appendix).
> * `For tasks other than Imagenet classification, it would be more convincing if the paper presents the comparison with some current baselines, in addition to the ablations`  At the largest scale we compare against numbers reported by other recent competitive works (Table 1). For completeness, we also include the original CLIP numbers (which are roughly comparable to the L/16 models from Figure 3/Table 4)
> 3. `Some acronyms might need further explanation such as the ones in Figure 3 for different models, e.g. B.16, S.16, …`
> * Thanks for pointing this out! You are correct - we have Table 4 in the appendix which defines the hyperparameters for each model, but we will amend the text, cite where these acronyms have been previously defined and point to the appendix for further information.
> 4. `No code is provided. `
> * Agreed - we hope to open source the code used to train models on LAION400M in Table 8 in time for the conference.
>
> (the final question will be addressed in the next comment)

---

> > ### Author Response · Authors · 2022-08-02
> > **Continuation of initial author response**
> >
> > Regarding the last question: `how do you identify the negative example`
> >
> > For a given example, we consider all other examples in the batch to be negatives.
> > In general, we have a batch of image and text pairs, for which we compute image representations $u_{1:N}$  and text representations $v_{1:N}$. Note $u_i$ and $v_i$ come from a single image-text pair (e.g. image and its caption). We want $u_i \cdot v_i$ to be high (aligned representations), but $u_i \cdot v_j$  to be low for every $j \neq i$.
> >
> > This is accomplished via the contrastive loss, which consists of two components as described in Equation 1 (Section 2.1). In the image-to-text component, every other image in the batch is a negative. In the text-to-image component, every other text in the batch is a negative. This formulation follows prior works, but we see that this loss could be noisy (if other elements in the batch are actually quite similar, but we encourage alignment to be low), and could possibly be improved by adapting methods e.g. from hard-negative sampling in other areas. Also, note this implies that the batch size may severely impact the learning process and its performance (higher batch size tends to be better, as we’ll have more negatives¹).
> >
> > ¹ Combined Scaling for Open-Vocabulary Image Classification, H. Pham, Z. Dai, G. Ghiasi, K. Kawaguchi, H. Liu, A. W. Yu, J. Yu, Y. Chen, M. Luong, Y. Wu, M. Tan, Q. V. Le, [arxiv:2111.10050](https://arxiv.org/abs/2111.10050)

---

> > > ### Comment · Reviewer_g9Jj · 2022-08-08
> > > **Response to authors for their first responses**
> > >
> > > I have read the reviews and responses from all reviewers and personally thank the authors for the clarifications and answers to my individual questions. I find them satisfactory and increase 1 point to support this paper for that reason.

---

### Official Review · Reviewer_Ycx8 · 2022-07-12

**Rating:** 8
**Confidence:** 4
**Soundness:** 4 excellent
**Presentation:** 4 excellent
**Contribution:** 3 good

**Summary:**

Along the line of mixute of experts that add expert layers to the Transformer architecture (e.g., ST-MoE in NLP and Vision MoE for vision), this paper introduces Language+Image MoE. Similar to other MoE, the single FeedForwardNetwork is replaced by an expert layer that contains many parallel FFNs, each of which is an expert. The difference is that given a sequence of tokens to process, a simple router learns to predict which experts should handle which tokens.

**Questions:**

> 1) Why In an imbalanced context all of the tokens from the minority modality get assigned to a single expert?
>2) some effects such as the behavior of global entropy threshold (text and image) are attributed to the imbalance dataset. How different would it be in the case of a balanced dataset?

**Ethics Review Area:**

["I don’t know"]

**Limitations:**

-

**Strengths And Weaknesses:**

Pros:
- Model size can increase while keeping computational cost constant (compared to BASIC the number of params is almost half for LIMoE H/14, while number of params per token- for inference- is less than 50%)
- As a sparse model, LIMoE avoids negative interference and catastrophic forgetting.
- Proposed new auxiliary loss(local/global entropy loss) and routing prioritization to prevent sending all tokens [from different modalities] to the same expert.
- Studing the scaling of LIMoE and ablations for various design decisions, e.g. loss, router architecture, number of experts, etc.

---

> ### Author Response · Authors · 2022-08-02
> **Initial author response**
>
> Many thanks for the time spent understanding and reviewing our paper! We’ve attempted to answer the questions below; please let us know if something doesn’t make sense, or if there’s something we can amend in the paper to make these clearer in the text.
>
>
> 1. `Why in an imbalanced context all of the tokens from the minority modality get assigned to a single expert?`
>
>
> * We discussed this a little bit in Section 2.2.1. In general, rather than necessarily a single expert, we observed that minority-modality tokens tend to choose a very small number of experts - fewer than their “fair share” of experts according to the token distributions. For example, with 64 experts and an image:text ratio of 12:1 (roughly matching the B/16 setup), one might expect $64*\frac{1}{12 + 1} \approx 5$ text experts. In practice, we would see only 1 or 2 experts being used for text. Note that this was empirically the case - as for why exactly this happens, it is as of yet unclear, and should definitely be further studied.
>
> * Assigning all tokens of the minority modality to a few experts  isn’t necessarily catastrophic if the experts have a large enough buffer size to process them. However, what we further noticed (Figure 2), is that the preferences of the minority modality tend to be quite unstable - e.g. at the start of training there will be 1 or 2 clear “text” experts, and midway through training, they suddenly become image experts and some other experts start processing a lot of text. This sudden switch hurts the training/validation performance. Alongside examining why this extreme per-modality expert assignment occurs, further study should be done to understand why these assignments are unstable without the right aux losses.
>
>
> 2. `Some effects such as the behavior of global entropy threshold (text and image) are attributed to the imbalance dataset. How different would it be in the case of a balanced dataset?`
>
> * We explored this somewhat in Figure 4, where we adjusted image sequence length by varying the number of patches per image. Our default setup has more image than text tokens (for B/16, image:text ratio is 12.2:1). In Figure 4, even when the ratio is 1:1 or there is more text than image, we find that the commonly used setup does not work, whereas our proposed training setup with BPR + auxiliary losses trains consistently stable models. Thus, we do not fully attribute the auxiliary losses’ functionality to modality imbalance.
>
> * In Section 2.2.2 and Figure 2 we break down what the losses are accomplishing. For example, they serve to “stabilize” the learned expert assignments. For instance, with the local loss, if a given expert learns to process 70% images and 30% text, it will continue with this balance and not suddenly change. This is a useful behavior independent of the ratio of image:text tokens.

---

### Meta-Review · Area_Chair_UBMV · 2022-08-29

**Recommendation:** Accept
**Confidence:** Certain

**Metareview:**

The authors use a mixture-of-experts model in a multimodal setting.  the reviewers consider the work technically strong and interesting; the AC concurs.

**Award:**

No

---

### Decision · Program_Chairs · 2022-09-14

Accept